# Transcriptome maps of general eukaryotic RNA degradation factors

**Salma Sohrabi-Jahromi[1†], Katharina B Hofmann[2†], Andrea Boltendahl[2], Christian Roth[1], Saskia Gressel[2], Carlo Baejen[2], Johannes Soeding[1]\*, Patrick Cramer[2]\***

[1]Quantitative and Computational Biology, Max-Planck-Institute for Biophysical Chemistry, Göttingen, Germany; [2]Department of Molecular Biology, Max-Planck-Institute for Biophysical Chemistry, Göttingen, Germany

**Abstract** RNA degradation pathways enable RNA processing, the regulation of RNA levels, and the surveillance of aberrant or poorly functional RNAs in cells. Here we provide transcriptome-wide RNA-binding profiles of 30 general RNA degradation factors in the yeast *Saccharomyces cerevisiae*. The profiles reveal the distribution of degradation factors between different RNA classes. They are consistent with the canonical degradation pathway for closed-loop forming mRNAs after deadenylation. Modeling based on mRNA half-lives suggests that most degradation factors bind intact mRNAs, whereas decapping factors are recruited only for mRNA degradation, consistent with decapping being a rate-limiting step. Decapping factors preferentially bind mRNAs with non-optimal codons, consistent with rapid degradation of inefficiently translated mRNAs. Global analysis suggests that the nuclear surveillance machinery, including the complexes Nrd1/Nab3 and TRAMP4, targets aberrant nuclear RNAs and processes snoRNAs.
DOI: https://doi.org/10.7554/eLife.47040.001

**\*For correspondence:**
johannes.soeding@mpibpc.mpg.de (JS);
patrick.cramer@mpibpc.mpg.de (PC)

[†]These authors contributed equally to this work

**Competing interests:** The authors declare that no competing interests exist.

## Introduction

The abundance of the different eukaryotic RNA species controls cell type and cell fate, and is determined by the balance between RNA synthesis and RNA degradation. Multiple mechanisms exist for RNA degradation (*Parker, 2012*). RNAs are generally exported to the cytoplasm, where they are degraded with an RNA-specific rate. Such canonical turnover is critical for RNA homeostasis. However, RNAs that are defective with respect to their processing, folding, assembly into RNA-protein particles, or their ability to be translated, are identified and rapidly degraded by surveillance pathways. Surveillance occurs both in the nucleus (*Schmid and Jensen, 2018*) and in the cytoplasm (*Zinder and Lima, 2017*). In the nucleus, aberrant RNAs resulting from upstream antisense transcription are quickly degraded. Moreover, some non-coding RNAs (ncRNAs) require processing by the degradation machinery (*Houseley et al., 2006*). Cytosolic RNAs also vary in their life-time, with mRNAs encoding cell cycle regulators or transcription factors having reported life-times in the range of minutes (*Geisberg et al., 2014*; *Miller et al., 2011*), whereas ribosomal RNAs live for days (*Turowski and Tollervey, 2015*). Therefore, RNA degradation kinetics need to be actively regulated to ensure the optimal life time for each transcript.

RNA degradation can occur from both ends of a transcript, and these processes are often coupled. During canonical mRNA turnover, degradation is thought to be initiated by shortening of the 3′ poly-adenylated (polyA) tail through two major deadenylation complexes, the Pan2/Pan3 complex and the multi-subunit Ccr4/Not complex (Ccr4, Not1, Pop2, Caf40) (*Wolf and Passmore, 2014*). Specific mRNAs can recruit selected deadenylating factors after loss of the polyadenylate-binding protein (Pab1) that protects the mRNA from degradation on the 3′ end (*Finoux and Séraphin, 2006*; *Goldstrohm et al., 2006*; *Semotok et al., 2005*), but the choice of deadenylation pathway

---

**eLife digest** Cells contain a large group of DNA-like molecules called RNAs. While DNA stores and preserves information, RNA influences how cells use and regulate that information. As such, regulating the quantities of different RNAs is a key part of how cells survive, grow, adapt and respond to changes. For example, messenger RNAs (or mRNAs for short) carry genetic information from DNA which the cell reads to produce proteins. RNAs that are not needed can be degraded and removed from the cell by RNA degradation proteins.

Most RNA degradation proteins need to be able to bind to RNA in order to work. A technique called "photoactivatable ribonucleoside-enhanced crosslinking and immunoprecipitation", often shortened to PAR-CLIP, can detect these proteins on their targets. The PAR-CLIP technique irreversibly links RNA-binding proteins to RNA and then collects those proteins and their bound RNAs for analysis. As with DNA, the RNAs can be identified using genetic sequencing. Degradation often starts at RNA ends, where specialized structures protect the RNA from accidental damage.

Using PAR-CLIP, Sohrabi-Jahromi, Hofmann et al performed a detailed study of 30 RNA degradation proteins in the yeast *Saccharomyces cerevisiae*. The results highlight the specialization of different proteins to different groups of RNAs. One group of proteins, for example, remove the protective 'cap' structure at the start of RNAs. Those mRNAs that are not efficiently producing proteins attracted a lot of these cap-removing proteins. The findings also identify proteins involved in RNA degradation in the cell nucleus – the compartment that houses most of the cell's DNA.

Together these findings provide an extensive data resource for cell biologists. It offers many links between different RNAs and their degradation proteins. Understanding these key cellular processes helps to reveal more about the mechanisms underlying all of biology. It can also shed light on what happens when these processes fail and the diseases that may result.

DOI: https://doi.org/10.7554/eLife.47040.002

---

remains unclear. Studies have pointed out a direct link between translation termination and mRNA degradation (reviewed in *Huch and Nissan, 2014*), in particular deadenylation, which is dependent on Pab1 and Ccr4 (*Webster et al., 2018*). A proposed stepwise model for deadenylation suggests that first the average yeast polyA tail length of 90 nucleotides (nt) is reduced to 50 nt by the Pan2/Pan3 complex, before further shortening via the Ccr4/Not complex (*Beilharz and Preiss, 2007*; *Brown and Sachs, 1998*; *Tucker et al., 2001*). When the polyA tail reaches a length of 10–12 nt, the mRNA is decapped (*Chowdhury et al., 2007*; *Tharun and Parker, 2001*), or subjected to exosome catalyzed degradation (*Bonneau et al., 2009*).

The second step in mRNA degradation is thought to be the removal of the 5′ cap by the decapping complex (Dcp1, Dcp2, Dcs1). The cap protects the mRNA from degradation by the 5′→3′ exonuclease Xrn1, which requires a 5′ monophosphate at the terminal residue (*Stevens and Poole, 1995*). Decapping is highly regulated by decapping enhancers such as the DEAD box helicase Dhh1, Edc2 and Edc3. Different potential mechanisms to trigger decapping include interference with translation initiation factors, facilitated assembly of the decapping machinery, and stimulation of Dcp2 catalytic function. Assembly of the decapping machinery occurs mainly after shortening of the polyA tail, which triggers decapping complex formation on the deadenylated 3′ end of mRNA and opening of the mRNA closed-loop structure (*Caponigro and Parker, 1995*; *Morrissey et al., 1999*). In this closed-loop model, the 5′ and 3′ ends of the mRNA are thought to be in close proximity by forming a complex between translation initiation factors binding to the 5′ cap and Pab1 associated with the 3′ end, thereby contributing to mRNA expression regulation (*Vicens et al., 2018*; *Wells et al., 1998*).

An alternative pathway of mRNA degradation after deadenylation is 3′→5′ degradation by the exosome and its auxiliary factors (*Anderson and Parker, 1998*). The exosome is a multi-subunit complex that consists of 10 core factors, comprising six members of the RNase PH protein family (Rrp43, Rrp45, Rrp42, Mtr3, Rrp41, Rrp46), three small RNA-binding proteins (Csl4, Rrp40, Rrp4) (*Allmang et al., 1999*), and the Rrp44/Dis3 protein, which harbors an exonuclease and an endonuclease domain (*Lebreton and Séraphin, 2008*; *Schaeffer et al., 2009*). In addition to its functions in the cytoplasm, the exosome fulfills multiple roles in nuclear RNA processing and degradation

(*Lykke-Andersen et al., 2009*; *Ogami et al., 2018*) for which it is additionally bound by Rrp6, another 3′→5′ exonuclease, Rrp47, and Mpp6 (*Milligan et al., 2008*; *Mitchell et al., 2003*; *Synowsky et al., 2009*).

For RNA degradation by the exosome, RNA first passes through either the TRAMP or the Ski complex. TRAMP is a nuclear poly-adenylation complex (*Houseley and Tollervey, 2009*) that is involved in many of the RNA maturation and degradation processes and exists in two isoforms, TRAMP4 (Trf4, Air2 and Mtr4) and TRAMP5 (Trf5, Air1 and Mtr4). These complexes harbor a pA polymerase (Trf4 or Trf5), a zinc-knuckle putative RNA-binding protein (Air1 or Air2), and an RNA helicase (Mtr4). Defective nuclear RNAs are tagged with a short polyA tail by TRAMP, making them a more favorable substrate for the exosome core (*Vanácová et al., 2005*). The Ski complex is required for cytoplasmic exosomal degradation. The Ski7 protein is stably bound to the cytoplasmic exosome through the Ski4 subunit (*van Hoof et al., 2002*). The Ski2, Ski3, and Ski8 proteins form a subcomplex interacting with Ski7, which is required for 3′→5′ degradation of mRNAs (*Araki et al., 2001*; *Brown et al., 2000*; *Wang et al., 2005*). The Ski2 protein is an ATPase of the RNA helicase family that generates energy by ATP hydrolysis to unwind secondary structures and dissociate bound proteins to deliver the RNA to the exosome.

To degrade eukaryotic mRNAs that are defective in translation, cytoplasmic quality control mechanisms exist (*Doma and Parker, 2007*). Normal and aberrant mRNAs can be discriminated by the translation machinery, and translationally defective mRNAs are guided to a degradation pathway. mRNAs with aberrant translation termination due to a premature translation termination codon are subjected to nonsense-mediated decay (NMD) (*Losson and Lacroute, 1979*). Substrates for NMD are identified by the Upf1 protein interacting with the translation termination complex followed by binding of the Upf2 and Upf3 proteins, which enhances the helicase activity of Upf1 (*Baker and Parker, 2004*; *Chakrabarti et al., 2011*). During NMD, the mRNA can be subjected to enhanced deadenylation, deadenylation-independent decapping and rapid 3′→5′ degradation (*Cao and Parker, 2003*; *Mitchell and Tollervey, 2003*; *Muhlrad and Parker, 1994*).

The large variety of different RNA degradation factors poses the question how RNA degradation pathways are selected and whether the RNA sequence can influence this selection. Answering this question requires a systematic analysis of the RNA-binding profiles of the involved protein factors. Although several transcriptome profiles of the RNA degradation factors Xrn1, Rrp44, Csl4, Rrp41, Rrp6, Mtr4, Trf4, Air2 and Ski2 have been reported (*Delan-Forino et al., 2017*; *Milligan et al., 2016*; *Schneider et al., 2012*; *Tuck and Tollervey, 2013*), we lack transcriptome-wide binding profiles for components of the deadenylation, decapping, and NMD machineries, as well as other subunits of the exosome complex. Thus, the task of systematically analyzing the binding of subunits from all known factors involved in RNA degradation to a eukaryotic transcriptome ('transcriptome mapping') has not been accomplished thus far.

Here we used photoactivatable ribonucleoside-enhanced crosslinking and immunoprecipitation (PAR-CLIP) to systematically generate transcriptome-wide protein binding profiles for 30 general RNA degradation factors in the yeast *Saccharomyces cerevisiae* (*S. cerevisiae*). In-depth bioinformatic analysis and comparisons with previously reported PAR-CLIP data provide factor enrichment on different RNA classes and the binding behavior for mRNAs and their associated antisense transcripts. The results also give insights into how the various degradation complexes, and also different subunits in these complexes, may be involved in the degradation of different RNA species. Several conclusions can be drawn with respect to degradation pathway selection, new functions for known factors can be proposed, and several hypotheses emerge that can be tested in the future. Finally, our dataset provides a rich resource for future studies of eukaryotic RNA degradation pathways, mechanisms, and the integration of mRNA metabolism.

## Results

### Transcriptome maps for 30 RNA degradation factors

In order to get a better understanding of RNA processing and degradation in a eukaryotic cell, we measured transcriptome-wide binding locations of 30 RNA degradation factors involved in mRNA deadenylation, decapping, exosome-mediated degradation, and in RNA surveillance pathways including nuclear RNA surveillance and cytoplasmic nonsense-mediated decay (NMD) (*Figure 1A,B*).

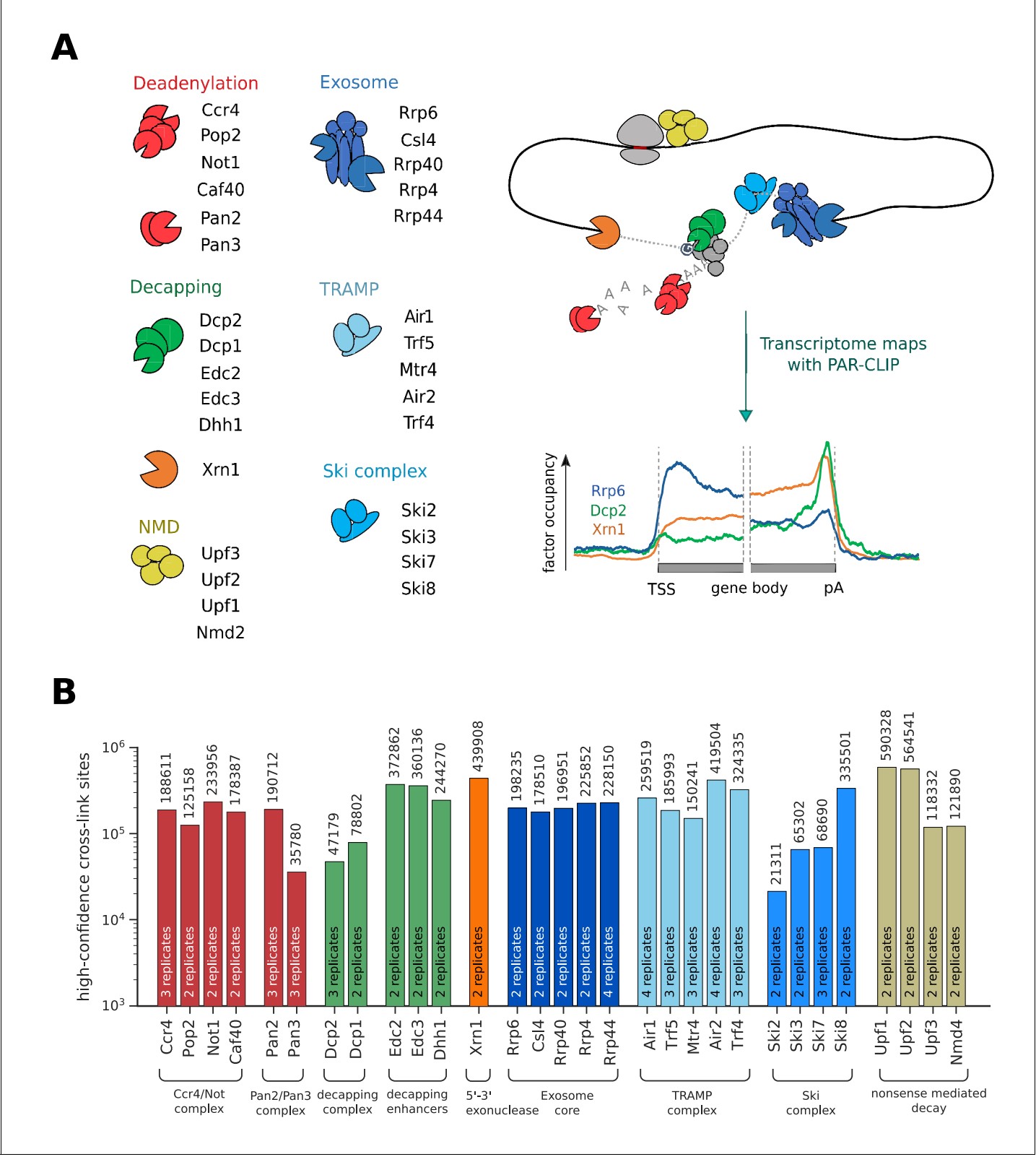

**Figure 1.** Overview of PAR-CLIP experiments performed in this study. (**A**) Overview of degradation pathways studied. (**B**) Number of high-confidence PAR-CLIP cross-link sites obtained for each factor after merging data from replicates.

DOI: https://doi.org/10.7554/eLife.47040.003

The following figure supplements are available for figure 1:

*Figure 1 continued*

**Figure supplement 1.** Biological replicate PAR-CLIP experiments have high correlation.
DOI: https://doi.org/10.7554/eLife.47040.004
**Figure supplement 2.** Western Blot analysis for all degradation factors analyzed in this study show IP efficiency.
DOI: https://doi.org/10.7554/eLife.47040.005

We performed PAR-CLIP in *S. cerevisiae* using our published protocol (*Battaglia et al., 2017*), with minor modifications (Materials and methods). The high reproducibility of these PAR-CLIP experiments is revealed by a comparison of two independent biological replicates that we collected for all 30 degradation factors (*Figure 1—figure supplement 1*), with Spearman correlations between 0.87 and 1.00 (mean: 0.94). We typically obtained tens of thousands of verified factor-RNA cross-link sites with p-values≤0.005 (*Figure 1B*). These transcriptome maps represent an extensive, high-confidence dataset of in vivo RNA-binding sites for factors involved in RNA degradation.

## Degradation factors exhibit transcript class specificity

We first compared degradation factor binding over different RNA classes. These included messenger RNA (mRNA), where we distinguished the 5′ untranslated region (5′ UTR), the coding sequence (CDS), introns, and the 3′ untranslated region (3′ UTR). We also included several classes of ncRNAs: ribosomal (r), transfer (t), small nucleolar (sno), and small nuclear (sn) RNAs, as well as stable unannotated transcripts (SUTs), cryptic unstable transcripts (CUTs), and Nrd1-unterminated transcripts (NUTs) (*Neil et al., 2009*; *Pelechano et al., 2013*; *Schulz et al., 2013*) (*Figure 2*).

A first analysis revealed that most PAR-CLIP sequencing reads fall into the mRNA transcript class, although many of the factors also show a considerable number of sequencing reads in ncRNAs, in particular rRNAs (*Figure 2A*). To obtain a more quantitative comparison, we defined log enrichment scores that reflect the preferences of factors in binding to a specific transcript class in comparison to other factors and classes. To correct for the different sizes of classes and different numbers of measured factor binding sites, we normalized the log enrichment scores by subtracting class- and factor-specific offsets, such that the mean for each class and each factor vanishes (*Figure 2B*, Materials and methods). This analysis highlights differences between degradation factors with respect to binding to various transcript classes, as will be discussed in detail below.

## RNA end-processing complexes differ in their targets

The catalytic subunit Pop2 and the core subunits Not1 and Caf40 of the deadenylase complex Ccr4/Not have similar binding preferences for the 5′ UTR, the CDS and 3′ UTR of mRNAs, for rRNAs, tRNAs, snoRNAs, and snRNAs (*Figure 2B*, highlighted in red). Compared to other deadenylation factors of the Ccr4/Not complex, the catalytically active subunit Ccr4 has different binding preferences, and is strongly enriched at mRNA introns. The second deadenylation complex, Pan2/Pan3, shows a similar binding preference as the Ccr4/Not complex (except for the Ccr4 subunit), consistent with its dominant role in yeast mRNA deadenylation (*Boeck et al., 1996*). Pan3 shows a strong binding preference for rRNAs and tRNAs.

For all decapping-related factors we observed similar binding preferences among each other (*Figure 2B*, highlighted in green). They show the strongest enrichment at SUTs and at mRNAs compared to the other transcript classes. Decapping factors bind preferentially to CDS and 3′ UTR as well as SUTs. This is consistent with previous findings that SUTs are degraded via Dcp2-dependent pathways in the cytoplasm (*Marquardt et al., 2011*; *Smith et al., 2014*; *Thompson and Parker, 2007*). Dcp2, which harbors the hydrolase activity that removes the 5′ cap, and the decapping activator Edc3, additionally bind to NUTs. The 5′ exonuclease Xrn1 shows a similar binding preference as the decapping factors (*Figure 2B*, highlighted in orange). Taken together, complexes and enzymes that are known to target mRNA ends for 3′ deadenylation and 5′ decapping and degradation show remarkably distinct binding specificities to different transcript classes.

## The exosome and surveillance factors

For the exosome we also observed binding to different RNA classes (*Figure 2B*, highlighted in royal blue). The core exosome subunits Csl4 and Rrp40 show similar cross-linking to rRNAs, tRNAs,

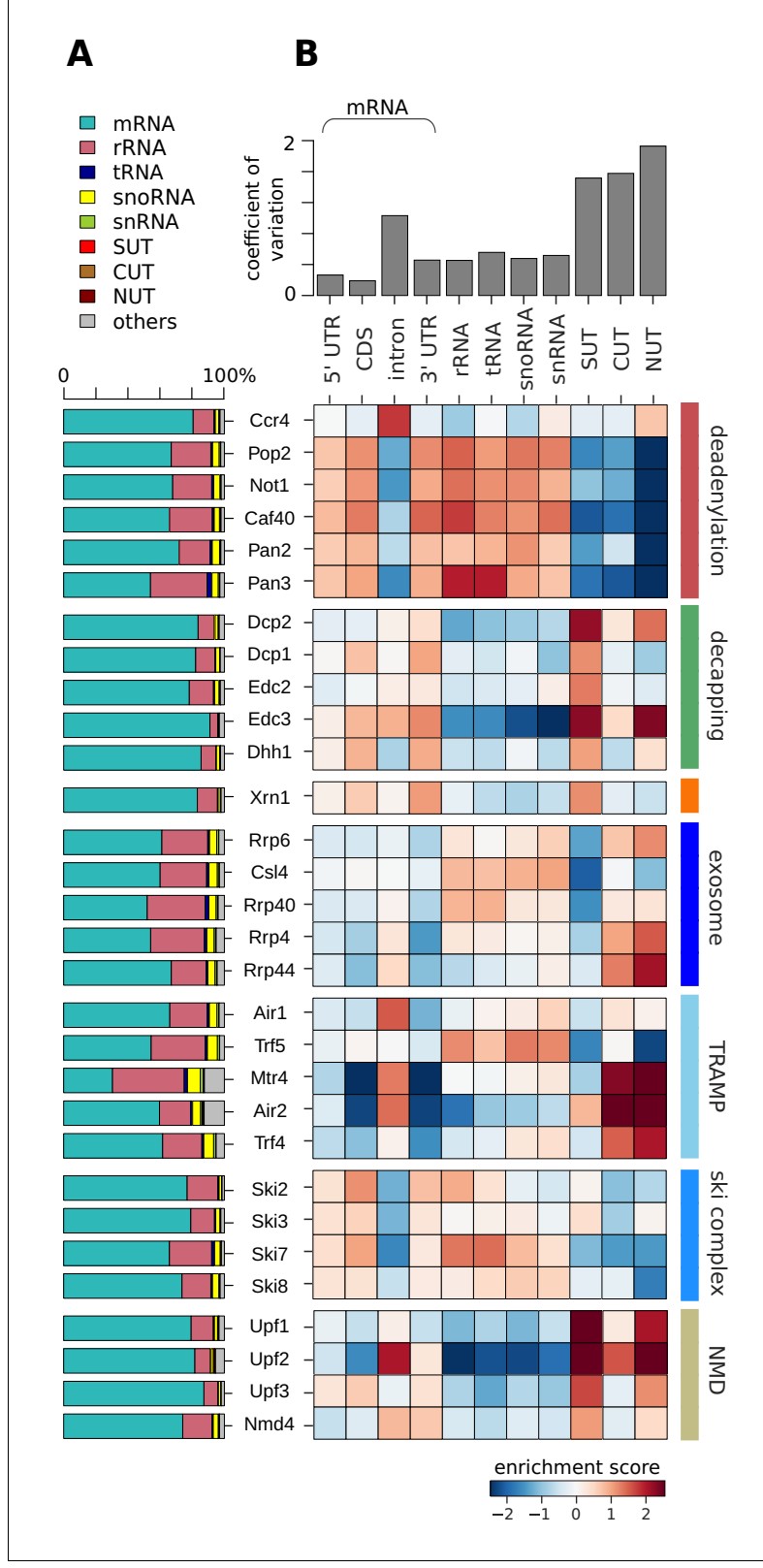

**Figure 2.** Distribution of degradation factor cross-link sites over the yeast transcriptome. (**A**) Fractions of high confidence PAR-CLIP sequencing reads of 30 yeast degradation factors fall into various transcript classes. Depicted classes are the following: messenger RNA (mRNA) in turquoise (n = 4,928), ribosomal RNA (rRNA) in antique pink (n = 24), transfer RNA (tRNA) in dark blue (n = 299), small nucleolar RNA (snoRNA) in yellow (n = 77),

*Figure 2 continued*

small nuclear RNA (snRNA) in green (n = 6), stable unannotated transcripts (SUTs) in red (n = 318), cryptic unstable transcripts (CUTs) in light brown (n = 637), Nrd1-unterminated transcripts (NUTs) in dark brown (n = 298)(Materials and methods). (B) Enrichment z-scores of high confidence PAR-CLIP cross-link sites of 30 yeast degradation factors (rows) in various segments of mRNA transcripts (left columns; UTR: untranslated region; intron; CDS: coding sequence), or other transcript classes as in A (other columns). The color-coded enrichment score shows the column and row normalized enrichment values of binding preferences of each factor for each transcript class (color encoded, depleted in blue and enriched in red). The coefficient of variation on top is the standard deviation divided by the mean for each transcript class.

DOI: https://doi.org/10.7554/eLife.47040.006

The following figure supplements are available for figure 2:

**Figure supplement 1.** Metagene profiles for subunits of the TRAMP complexes on snoRNA genes.
DOI: https://doi.org/10.7554/eLife.47040.007
**Figure supplement 2.** Different transcript classes have comparable U-content.
DOI: https://doi.org/10.7554/eLife.47040.008

---

snoRNAs, and snRNAs. The catalytic exosome subunit Rrp44 and the core subunit Rrp4 binds to introns of mRNAs, but preferentially to the short-lived, nuclear CUTs and NUTs. Rrp6, a subunit that is exclusively present in the nuclear exosome complex, shows binding to rRNAs, snoRNAs, snRNAs, CUTs and NUTs. This is consistent with the suggestion that the factor is needed for nuclear processing of such non-coding transcripts and degradation of short-lived nuclear transcripts (*Heo et al., 2013*; *Vasiljeva and Buratowski, 2006*). This complex distribution of cross-links for different exosome subunits to different RNA classes reflects the distinct functions of the exosome in nuclear RNA surveillance, processing of stable ncRNAs, and cytoplasmic mRNA degradation (*Zinder and Lima, 2017*).

The two TRAMP complexes TRAMP4 and TRAMP5 show clearly distinct cross-linking patterns (*Figure 2B*, highlighted in light blue). TRAMP4 subunits (Mtr4, Air2, Trf4) are enriched in introns, consistent with a function on mRNAs, and on SUTs, CUTs, and NUTs. The TRAMP5 complex (Mtr4, Air1, Trf5) shows binding enrichment for introns, rRNAs, tRNAs, snRNAs, and snoRNAs. This is in agreement with previous data, which showed rRNA binding for Mtr4 and exosome subunits (*Delan-Forino et al., 2017*; *Schneider and Tollervey, 2013*). Moreover, the TRAMP complex cooperates with the Nrd1/Nab3 complex and the nuclear exosome complex during the maturation and 3′ pre-processing of snoRNAs (*Grzechnik and Kufel, 2008*). To distinguish binding upon degradation and binding in order to pre-process snoRNAs, we investigated metagene profiles of TRAMP subunits along snoRNA genes (*Figure 2—figure supplement 1*). Air1/Trf5 bind almost exclusively to the gene body whereas Air2/Trf4 bind downstream of the 3′ end. This suggests that TRAMP5 is mainly involved in snoRNA degradation, whereas TRAMP4 may work together with the Nrd1/Nab3 machinery to pre-process snoRNAs (*Figure 2—figure supplement 1*) and to target NUTs, SUTs, and CUTs for degradation (*Figure 2B*, highlighted in light blue).

The cross-linking preferences of subunits of the Ski complex differ only slightly from each other (*Figure 2B*, highlighted in cyan). All Ski complex subunits bind the 5′ UTR, CDS, and 3′ UTR of mRNAs, rRNAs, tRNAs, snoRNAs, and snRNAs. The Ski2 subunit preferentially binds to the CDS of mRNAs, consistent with its function as a helicase to detach bound proteins from the mRNAs (*Houseley and Tollervey, 2009*; *Lebreton and Séraphin, 2008*). The exosome adaptor subunit Ski7 preferentially binds rRNAs and tRNAs. These patterns are consistent with the model that the exosome cooperates with distinct accessory complexes and factors to target different transcript classes. Finally, we observed similar cross-linking patterns for all NMD factors with strong binding to SUTs and NUTs (*Figure 2B*, highlighted in yellow). Upf2 shows an additional binding preference to introns and CUTs. Upf3 also binds to the 5′ UTR, CDS, and 3′ UTR of mRNAs, and Nmd4 binds to introns and 3′ UTRs of mRNAs.

## Distinct factor distribution along mRNA

We next focused on degradation factor distribution on mRNAs. We prepared metagene profiles showing the average occupancy of each factor around the mRNA transcription start sites (TSS) and the poly-adenylation (pA) sites, respectively (*Figure 3*). The Pan2/Pan3 deadenylase complex and

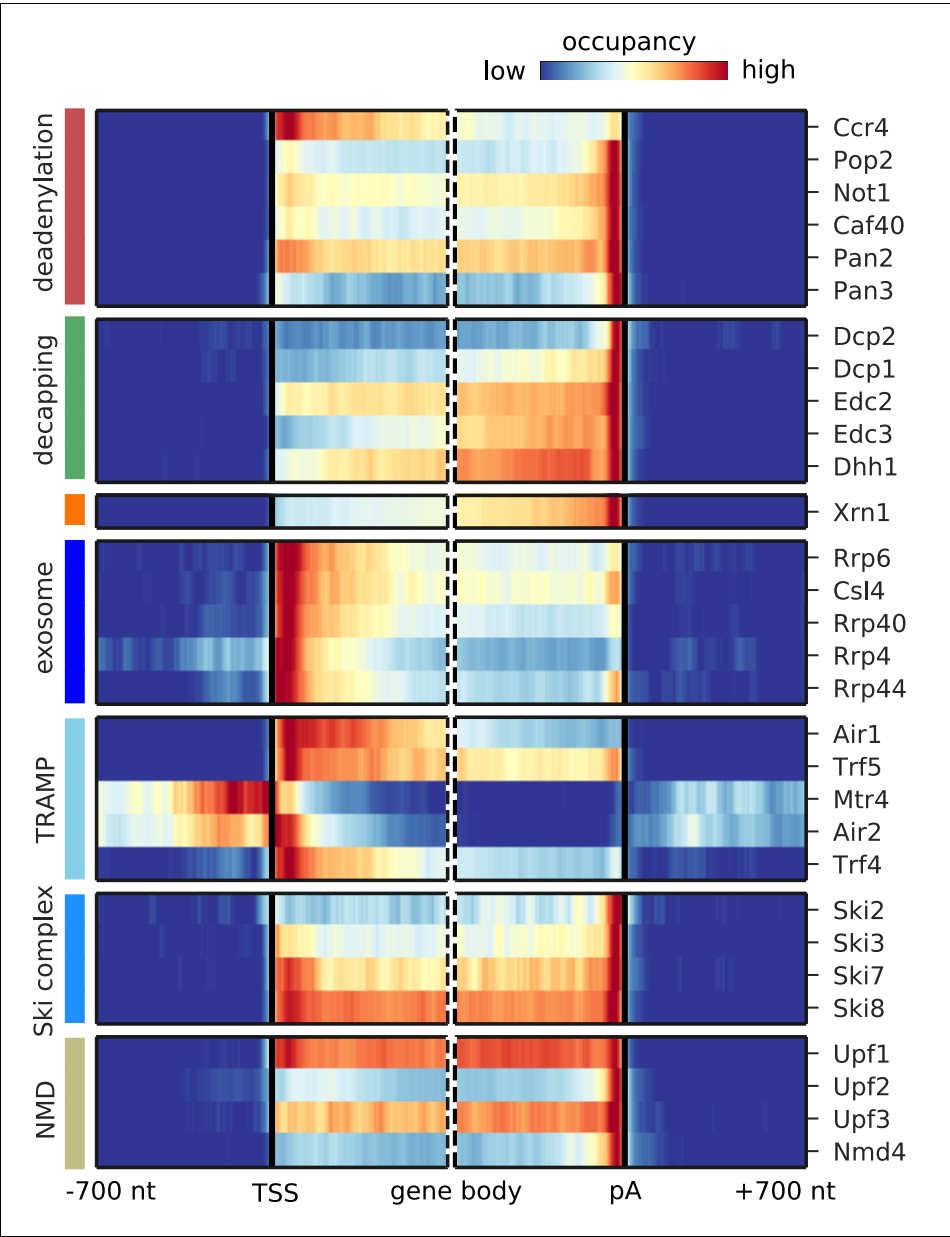

**Figure 3.** Metagene analysis of degradation factor binding on mRNAs. Averaged occupancy profiles of degradation factors over mRNAs aligned around their transcription start site (TSS) (n = 3,193, left) and around their poly-adenylation (pA) site (n = 3,193, right) in a window of [±700 nt]. Regions that have neighboring transcripts on the same strand were removed to avoid contaminating profiles (Materials and methods). Factors are grouped according to their functional role; from top to bottom: deadenylation, decapping, Xrn1, exosome, TRAMP complex, Ski complex, or NMD. The color code shows the average occupancy normalized between the minimum (blue) and maximum (red) values per profile.

DOI: https://doi.org/10.7554/eLife.47040.009

The following figure supplements are available for figure 3:

**Figure supplement 1.** Metagene profiles of yeast RNA degradation factors centered on translation start and stop sites in comparison to TIF-annotated TSS and pA sites.

DOI: https://doi.org/10.7554/eLife.47040.010

**Figure supplement 2.** Comparison of binding profiles on genes containing annotated upstream sense NUTs with all mRNAs.

DOI: https://doi.org/10.7554/eLife.47040.011

*Figure 3 continued on next page*

*Figure 3 continued*

**Figure supplement 3.** Metagene analysis of degradation factor binding on mRNAs after removing signals from known NUTs and CUTs.

DOI: https://doi.org/10.7554/eLife.47040.012

the Ccr4/Not subunits Pop2, Not1, and Caf40 all cross-link upstream of the 3′ end of mRNA with the highest enrichment at the pA site, as expected from their function in shortening the polyA tail. The catalytic subunit Ccr4 binds strongly in the 5′ region of mRNAs. All 5′ decapping factors bind upstream of the pA site, and all but the catalytically active subunit Dcp2 show increasing occupancy towards the 3′ end of mRNAs. These patterns can be explained if decapping factors are pre-bound to mRNAs that form a closed loop that holds the RNA ends in proximity. In contrast, Dcp2 binds almost exclusively at the pA site, suggesting that it might be recruited only upon active mRNA degradation. The cytoplasmic 5′ exonuclease Xrn1 has the highest occupancy towards the 3′ end, similar to the previously published crosslinking and cDNA analysis (CRAC) data (*Tuck and Tollervey, 2013*), thereby resembling the binding profiles of the decapping factors. Comparison of the binding profiles aligned at the pA site or alternatively with profiles aligned at the translation stop codon shows that the binding preference indeed lies at the end of the 3′ UTR independent of the stop codon position (*Figure 3—figure supplement 1B,C*).

The exosome core subunits (Csl4, Rrp40, and Rrp4) and the catalytically active subunits (nuclear: Rrp6, cytoplasmic: Rrp44) cross-link to the 5′ end of the transcript (*Figure 3*), possibly because the exosome binds to the 5′ end while digesting the 3′ end, or more likely because the exosome slows down towards the remaining 5′ end of mRNAs after rapid degradation from the 3′ end. Both TRAMP complexes bind mainly in the 5′ region of mRNAs near the TSS, as previously observed for Mtr4 and Trf4 (*Tuck and Tollervey, 2013*).

The Ski complex components Ski7 and Ski8 occupy the entire mRNA with increasing occupancy towards the pA site, whereas Ski2 and Ski3 show more discrete binding towards the polyA tail (*Figure 3*). The NMD factors Upf1 and Upf3 show binding over the entire mRNA with highest occupancy at the pA site, consistent with their role in scanning for premature stop codons in mRNAs and remodeling of the 3′ end of protein-RNA complexes and completion of mRNA decay (*Franks et al., 2010*). In addition, Upf2 and Nmd4 show strongest binding near the 3′ ends of mRNAs. Taken together, the distribution of cross-links along mRNA transcripts differs between degradation complexes and in some cases also between their subunits.

## Surveillance of aberrant nuclear ncRNA

Pervasive transcription of the genome leads to many short-lived aberrant RNAs that must be rapidly detected and degraded in the nucleus. We previously reported that the RNA surveillance factors Nrd1 and Nab3 strongly cross-link to aberrant upstream antisense RNA that stems from bidirectional transcription (*Schulz et al., 2013*). In order to find factors cross-linking to aberrant ncRNAs, we plotted the occupancy of all 30 investigated factors on the antisense strand of known mRNAs (*Figure 4*). For comparison, we plotted the published Nrd1 and Nab3 profiles in the first two lanes of *Figure 4*. The factors are involved in processing and degradation of Nrd1-unterminated transcripts, or NUTs (*Schulz et al., 2013*), and are expected to show similar binding to upstream antisense RNA as Nrd1 and Nab3. Indeed, we observed a similar binding pattern for all exosome subunits (Rrp6, Csl4, Rrp40, Rrp4, Rrp44) and subunits of the TRAMP4 complex (Mtr4, Air2, Trf4). Consistent with this, these factors also bind strongly to previously annotated NUTs and CUTs (*Figure 2*) and show strong enrichment of Nrd1 and Nab3 motifs (GTAG, CTTG) around their cross-link sites (*Figure 4—figure supplement 1*).

It has been shown that Nrd1 is involved in terminating transcripts upstream of the TSS. We also observe a strong signal for binding upstream of TSS on the sense strand for Air2 and Mtr4 (*Figure 3*). This suggests that the TRAMP4 complex is involved in degradation of those Nrd1-regulated upstream sense transcripts. To investigate this hypothesis, we compared the binding profiles around the TSS of 459 protein-coding genes, previously annotated as having upstream Nrd1-unterminated transcripts, or NUTs (*Schulz et al., 2013*), with the profiles obtained for all mRNAs (*Figure 3—figure supplement 2A,B*). TRAMP4 and the exosome subunits show a strong preference for binding to the upstream promoter region of genes that are controlled by the Nrd1/Nab3 complex. To further

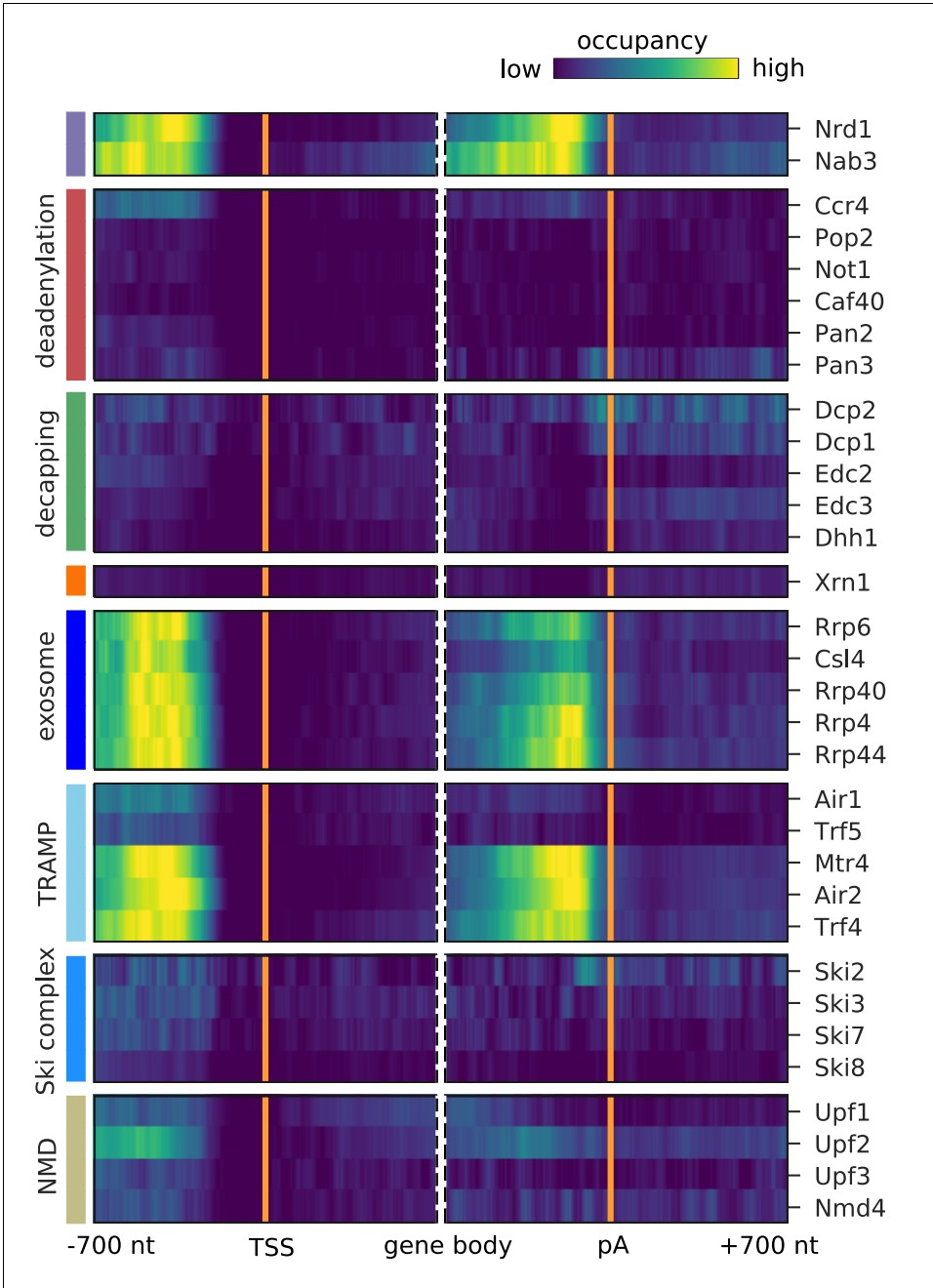

**Figure 4.** Surveillance of aberrant nuclear antisense RNAs by the exosome and the TRAMP4 complex. Averaged occupancy profiles of degradation factors binding to transcripts antisense of mRNAs aligned around transcription start site (TSS) (n = 3,076, left) and around their poly-adenylation (pA) site (n = 2,705, right) in a window of [±700 nt]. Regions with annotated genes on the antisense strand are removed to avoid contaminating the profiles (Materials and methods). The color code shows the average occupancy normalized between the minimum (blue) and maximum (yellow) values per profile. On top, previously published PAR-CLIP profiles for Nrd1 and Nab3 are included for comparison (*Schulz et al., 2013*).
DOI: https://doi.org/10.7554/eLife.47040.013

The following figure supplements are available for figure 4:

**Figure supplement 1.** Motif enrichment analysis shows enrichment of Nrd1/Nab3 motifs for the TRAMP4 and the exosome complex.
DOI: https://doi.org/10.7554/eLife.47040.014

*Figure 4 continued on next page*

*Figure 4 continued*

**Figure supplement 2.** The aberrant nuclear ncRNAs bound by components of the exosome and the TRAMP4 complex are primarily NUTs and CUTs.
DOI: https://doi.org/10.7554/eLife.47040.015

confirm that this upstream signal originates from NUTs and CUTs, we excluded cross-link sites that fall within such previously annotated regions. We then compared the binding profiles generated from the remaining binding sites on mRNAs (*Figure 3—figure supplement 3*). Upon filtering, the signal upstream of the TSS for Air2 and Mtr4 decreases, showing that Nrd1-mediated regulation is the primary cause for this upstream signal. Comparison of the observed antisense profiles (*Figure 4*) with those obtained after excluding cross-link sites in previously annotated NUT and CUT regions (*Figure 4—figure supplement 2*) confirms that most of the signal originates from transcripts that are targeted by the Nrd1/Nab3 machinery.

These results are consistent with the idea that the nuclear RNA surveillance machinery involves, in addition to Nrd1 and Nab3, the TRAMP4 complex and the nuclear exosome. Indeed, it was reported that TRAMP4 can add a short polyA tail on aberrant RNAs (*Wyers et al., 2005*), which may trigger degradation by the nuclear exosome. It was also recently shown that Nrd1 and Trf4 interact, providing a basis for coupling surveillance-mediated termination to RNA degradation (*Tudek et al., 2014*).

## Interactions between RNA processing machineries

To find out which groups of factors can work together in degrading transcripts, we analyzed their tendency to co-occupy the same transcripts by calculating the Pearson correlation of their occupancy across all transcripts (*Figure 5A*). We also analyzed their co-localization, that is the tendency of a factor to bind near to another factor's binding sites using a range of ±40 nt from each cross-link site (*Figure 5B*). To relate these profiles to those of other factors, we included previously published PAR-CLIP profiles from our lab (*Supplementary file 1*). Profiles were available for factors that function in nuclear RNA surveillance (Nrd1, Nab3), cap binding (Cbc2), mRNA transcript elongation (Bur1, Bur2, Ctk1, Ctk2, Cdc73, Ctr9, Leo1, Paf1, Rtf1, Set1, Set2, Dot1, Spt5, Spt6, Rpb1), pre-mRNA splicing (Ist3, Nam8, Mud1, Snp1, Luc7, Mud2, Msl5), pre-mRNA 3′ processing (Pab1, Pub1, Rna15, Mpe1, Cft2; Yth1), transcription termination (Rat1, Rai1, Rtt103, Pcf11), and mRNA export (Hrp1, Tho2, Gbp2, Hrb1, Mex67, Sub2, Yra1, Nab2, Npl3) (*Baejen et al., 2017*; *Baejen et al., 2014*; *Battaglia et al., 2017*; *Schulz et al., 2013*).

Co-occupancy and co-localization plots for all factors can be found in *Figure 5—figure supplements 1* and *2*, respectively. A two-dimensional embedding of co-occupancy profiles between all these processing factors is shown in *Figure 5C*. It represents the degree of similarities between co-occupancy of transcripts (*Figure 5A*) in terms of the distance in two dimensions. The two-dimensional embedding of the co-localization matrix in *Figure 5B* shows a similar clustering (*Figure 5—figure supplement 3*). This extensive global analysis suggests which factors reside in functional complexes and which functional complexes may interact during RNA processing and degradation. The analysis recovers several established interactions between subunits of known complexes and between different complexes, providing a positive control. For example, all factors of the decapping complex show very high co-occupancy and co-localization, as do Air2 and Mtr4, which reside in the TRAMP4 complex.

The analysis contains a lot of new information, forcing us to focus here on a few interesting, novel findings (*Figure 5C*). First, the largest cluster is formed by the previously analyzed factors involved in transcription elongation by RNA polymerase II (cluster 1) and in co-transcriptional pre-mRNA processing, including cap-binding complex (Cbc2), 3′ processing, transcription termination, and RNA export. The degradation factors Ccr4 and Air1 also reside in this cluster, maybe reflecting the role of Ccr4 in transcription elongation (*Kruk et al., 2011*). A second cluster is formed by splicing factors (cluster 2). Factors involved in nuclear and cytoplasmic exosomal degradation (Rrp6, Csl4, Rrp4, Rrp40 and Rrp44) form a third cluster (cluster 3). Close to cluster 3, we find the TRAMP4 complex subunit Trf4, the elongation factors Dot1, Paf1, Leo1, and the termination factors Pcf11 and Rai1. Rai1 has been shown to detect and remove incomplete 5′ cap structures, to subject aberrant pre-mRNAs to nuclear degradation (*Jiao et al., 2010*).

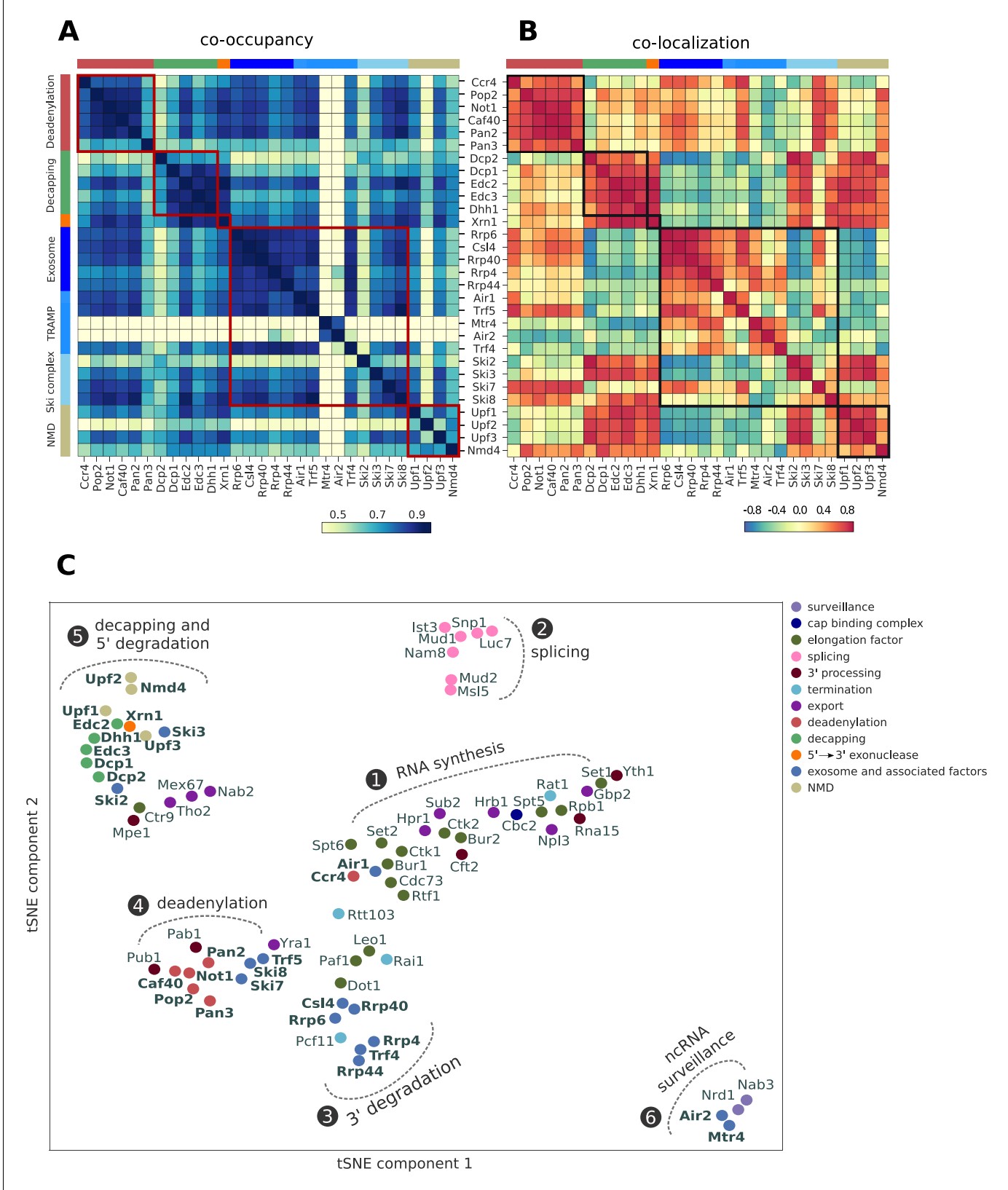

**Figure 5.** Global co-occupancy and co-localization analysis reveals unexpected cooperation between factors from different complexes and pathways. (**A**) Matrix of pairwise correlation coefficients of factor occupancies evaluated over all transcripts. (**B**) Matrix of co-localization based on the enrichment of factor x binding within 40 nt of the cross-link site of factor x′ (Materials and methods). (**C**) Two-dimensional embedding of the co-occupancies in (**A**)

*Figure 5 continued on next page*

*Figure 5 continued*

analyzed for 74 RNA processing factors with tSNE, including 30 factors from this study (highlighted in bold), and 44 factors from previous studies (*Baejen et al., 2017*; *Baejen et al., 2014*; *Battaglia et al., 2017*; *Schulz et al., 2013*) (*Supplementary file 1*). Factors that are plotted in close proximity show a preference for binding to the same transcripts. Clusters present factors involved in RNA synthesis (1), splicing (2), 3′ processing (3), deadenylation (4), decapping (5), nuclear ncRNA processing (6), and surveillance (7).

DOI: https://doi.org/10.7554/eLife.47040.016

The following figure supplements are available for figure 5:

**Figure supplement 1.** Co-occupancy for 74 RNA processing factors.

DOI: https://doi.org/10.7554/eLife.47040.017

**Figure supplement 2.** Co-localization coefficients for all 74 RNA processing factors.

DOI: https://doi.org/10.7554/eLife.47040.018

**Figure supplement 3.** Two-dimensional embedding of co-localization between 74 RNA processing factors.

DOI: https://doi.org/10.7554/eLife.47040.019

A fourth cluster is formed by mRNA deadenylation factors together with polyA tail binding proteins (Pab1 and Pub1), Ski7, Ski8, Trf5, and the export factor Yra1 (cluster 4). This is consistent with coupled mRNA deadenylation and subsequent degradation from its 3′ end by the exosome with the Ski or TRAMP complex as adaptors. The fifth cluster is formed by mRNA decapping factors, which cluster together with Xrn1, suggesting a coupling of mRNA decapping with degradation from the 5′ end by Xrn1 (cluster 5). The NMD-involved factors Upf1, Upf2, Upf3 and Nmd4, and Ski2 and Ski3 are also found in cluster 5. The high correlation between Xrn1 and Ski2 has been reported in a CRAC experiment (*Tuck and Tollervey, 2013*). The elongation factor Ctr9, the 3′ processing factor Mpe1 and the export factors Tho2, Mex67 and Nab2 are also found in cluster 5. A last cluster (cluster 6) is formed by factors involved in nuclear RNA surveillance, including Air2, Mtr4 and the Nrd1/Nab3 complex. Taken together, these findings are consistent with known functional associations and physical interactions between factors and suggest intriguing new associations to be investigated in future work.

## 5′ degradation machinery senses translation efficiency

To study the link between cytosolic mRNA translation and degradation, we compared the occupancy of degradation factors on mRNAs to their average codon-optimality score ('transcript optimality') (*Figure 6A*, *Figure 6—figure supplements 2–8A*). We found that the 5′ decapping machinery and Xrn1 preferentially bind transcripts with low transcript optimality. In contrast, the 3′ deadenylation machinery and the exosome bind more strongly to optimal transcripts. We asked whether this correlation with codon optimality is introduced by only a few differentially bound codons or by global enrichment/depletion of optimal codons. For this purpose, we introduced a 'codon enrichment score', which measures a codon's enrichment in the set of transcripts bound by the factor relative to the yeast mRNA pool. For Dcp2 this enrichment score is high on non-optimal codons, and low on optimal codons, whereas the opposite trend is observed for Ccr4 and most degradation factors (*Figure 6B*, *Figure 6—figure supplement 1–7*) This is consistent with a model that ribosome stalling on translationally inefficient codons can lead to recruitment of Dcp2 and Xrn1 and subsequent 5′ degradation of the transcript (*Heck and Wilusz, 2018*).

To investigate the significance of the correlation between transcript optimality and binding of the 5′ degradation machinery, we compared the contribution of several mRNA features in explaining the occupancy patterns retrieved from PAR-CLIP experiments. Since mRNA expression, half-life, and translation optimality are inter-correlated (*Figure 6—figure supplement 8*), a causative effect of one of these features on binding strength may lead to correlations with all three features. To better distinguish correlation from causation, we used linear regression analysis to explore whether correlations between factor binding and optimality are better explained with other mRNA features (*Figure 6—figure supplement 9*). We assessed the significance of features via the likelihood ratio test on the multi-variate linear regression model for occupancy. The likelihood ratio test calculates the significance of a feature from the change of the likelihood (quantifying the prediction quality) upon removal of that feature from the regression model. For decapping enhancers (Edc2, Edc3, and Dhh1) and Xrn1, low codon optimality is the most determining feature for binding (*Figure 6C*). The same is true for NMD factors Upf1 and Upf3, which are known to bind non-optimal transcripts

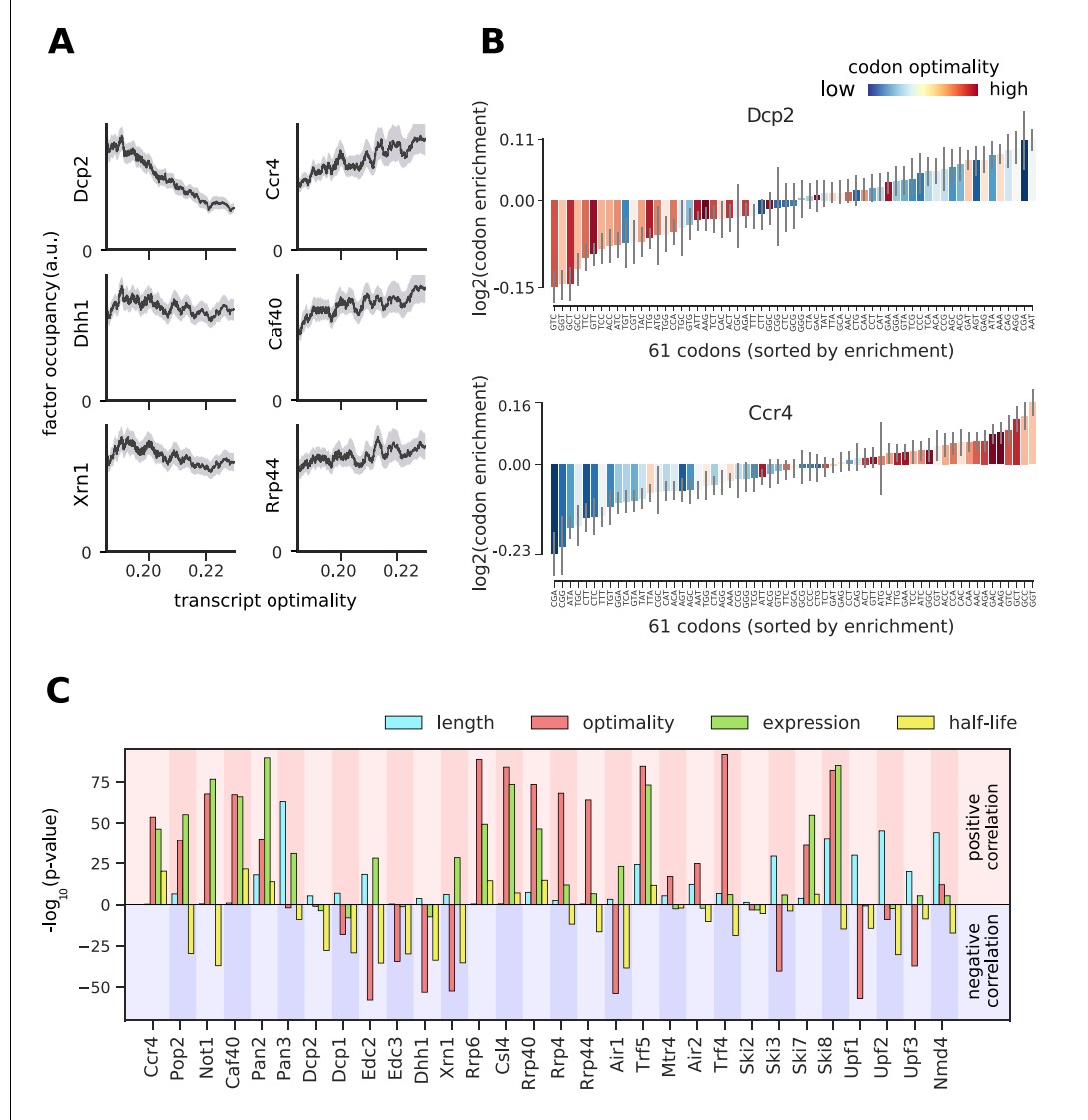

**Figure 6.** Binding preferences reveal a link between decapping-mediated degradation and translation. (**A**) Total occupancy per mRNA (according to TIF-seq annotation) for six factors as a function of the average mRNA codon optimality (transcript optimality). The occupancy of factors from the 5′→3′ degradation machinery (decapping and Xrn1, left) decreases with increasing transcript optimality, whereas the occupancy of factors from the 3′→5′ degradation machinery (Ccr4 and Caf40 deadenylation complex subunits and exosome subunit Rrp44, right) increases with increasing average codon optimality. (Gray shading: 95% confidence intervals generated by bootstrapping mRNAs). (**B**) Codon enrichment in transcripts bound by Dcp2 and Ccr4 compared to the average frequency over all mRNAs. The bar colors represent codon optimality, with highly optimal codons shown in dark red. (Thin gray lines: 90% confidence intervals generated by bootstrapping coding sequences.) (**C**) Significance of correlations between the binding strength of degradation factors and transcript length, transcript optimality (*Pechmann and Frydman, 2013*), expression level (*Baejen et al., 2017*), and half-life derived by multivariate linear regression analysis (Materials and methods). Bars are separated according to the direction of correlation with positive correlation marked by a red background and negative correlation marked by a blue background.
DOI: https://doi.org/10.7554/eLife.47040.020

The following figure supplements are available for figure 6:

**Figure supplement 1.** Occupancies of deadenylation factors (Ccr4, Pop2, Not1, Caf40, Pan2, and Pan3) compared to transcript length, optimality, expression level, and half-life.
DOI: https://doi.org/10.7554/eLife.47040.021

**Figure supplement 2.** Occupancies of decapping factors (Dcp2, Dcp1, Edc2, Edc3, and Dhh1) compared to transcript length, optimality, expression level, and half-life.
DOI: https://doi.org/10.7554/eLife.47040.022

**Figure supplement 3.** Occupancy of Xrn1 compared to transcript length, optimality, expression level, and half-life.
DOI: https://doi.org/10.7554/eLife.47040.023

*Figure 6 continued on next page*

*Figure 6 continued*

**Figure supplement 4.** Occupancies of exosome components (Rrp6, Csl4, Rrp40, Rrp4, and Rrp44) compared to transcript length, optimality, expression level, and half-life.

DOI: https://doi.org/10.7554/eLife.47040.024

**Figure supplement 5.** Occupancies for components of the TRAMP complex (Air1, Trf5, Mtr4, Air2, and Trf4) compared to transcript length, optimality, expression level, and half-life.

DOI: https://doi.org/10.7554/eLife.47040.025

**Figure supplement 6.** Occupancies for components of the Ski complex (Ski2, Ski3, Ski7, and Ski8) compared to transcript length, optimality, expression level, and half-life.

DOI: https://doi.org/10.7554/eLife.47040.026

**Figure supplement 7.** Occupancies for components of the NMD pathway (Upf1, Upf2, Upf3, and Nmd4) compared to transcript length, optimality, expression level, and half-life.

DOI: https://doi.org/10.7554/eLife.47040.027

**Figure supplement 8.** Distributions of transcript length, half-life, expression level and transcript optimality for yeast mRNAs.

DOI: https://doi.org/10.7554/eLife.47040.028

**Figure supplement 9.** Correlation between binding to degradation factors and transcript length, codon-optimality, expression, and half-life.

DOI: https://doi.org/10.7554/eLife.47040.029

(*Celik et al., 2017*). This result confirms the importance of the translation efficiency for the stability of cytosolic mRNAs and strengthens our finding that transcripts with low average codon optimality are preferentially targeted by the decapping machinery and degraded from the 5′ end.

## Decapping factors are enriched upon RNA degradation

Although decapping occurs at the 5′ end of mRNAs, decapping factors show a strong occupancy near the 3′ end (*Figure 3*). To investigate this further, we compared metagene profiles of decapping factors between stable (top 25%) and unstable (bottom 25%) transcripts, using mRNA half-life estimates (*Figure 7A*, Materials and methods). On both stable and unstable mRNAs, Dcp1, Edc2, Edc3, and Dhh1 show increased binding near the 3′ end, but unstable RNAs show a higher occupancy in the transcript body. The catalytically active subunit Dcp2 binds almost exclusively at the 3′ end and has a higher occupancy on unstable transcripts. Moreover, A-rich 4-mers are abundant around the proximity (eight nt) of Dcp2-cross-link sites (*Figure 7C*), indicating a binding preference of Dcp2 for A-rich RNA sequences. Overall, these binding patterns suggest that decapping factors are bound in transcript bodies and near the 3′ end of transcripts, and that through closed-loop formation of the mRNA they are in close proximity to the 5′ end. Decapping factors might also travel with the 5′→3′ exonuclease Xrn1 upon RNA degradation.

Decapping factors may bind to complete mRNAs or to transcripts that are in the process of being degraded. To quantify these two behaviors, we combined our PAR-CLIP occupancy data with RNA half-life estimates (Materials and methods). We modeled the occupancy of factors on mRNA as the sum of binding to all transcripts ($b$) and surplus binding to transcripts that are in the process of degradation ($\frac{a}{t_{1/2}}$). Therefore, we can model occupancy as a function of half-life with a linear equation ($\text{occupancy} = \frac{a}{t_{1/2}} + b$). In cases where there is no surplus binding upon active degradation, that is the occupancy is the same as in intact RNAs, 'a' will be zero. For 5′ decapping factors, this model closely fits the occupancy patterns retrieved from our experiments (*Figure 7B*), other degradation factors also follow this pattern to varying degrees (*Figure 6—figure supplements 1–7*). In particular, Dcp2 shows a very high a/b ratio, revealing that it cross-links preferentially to transcripts that are being degraded. This analysis strongly suggests that the 5′ decapping machinery, although present to some extent on complete mRNAs, is enriched when mRNAs are degraded.

## Discussion

Here we report transcriptome-wide binding maps for 30 RNA degradation factors in yeast. A detailed bioinformatics analysis of these maps revealed how degradation factors vary in their binding specificities for different classes of RNAs and with respect to their preferred locations on RNA transcripts. Global comparisons of the profiles alongside previously published profiles of other RNA-binding factors revealed clusters of factors that co-occupy RNAs or co-localize on RNAs. Our data

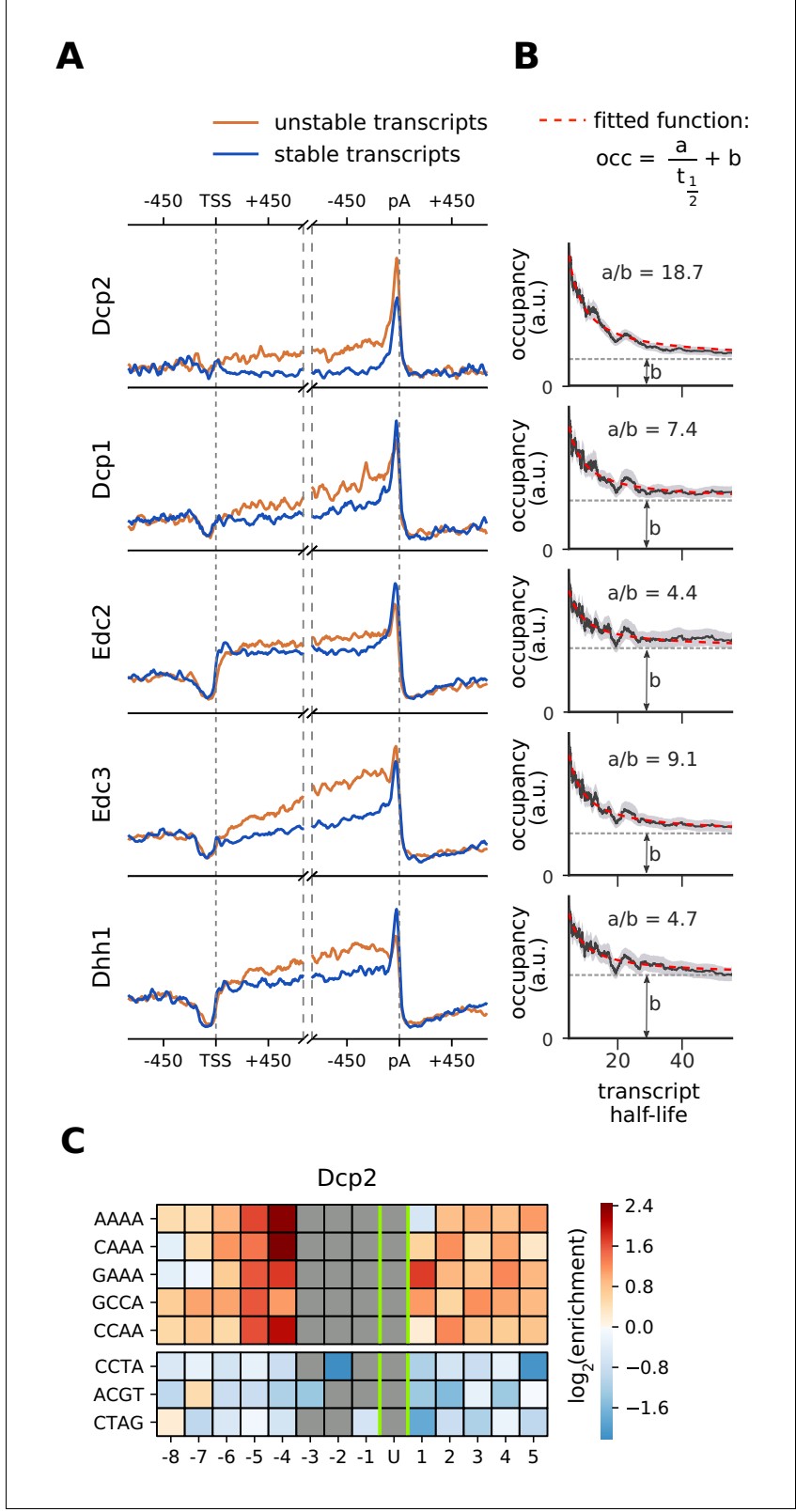

**Figure 7.** Location and recruitment of the decapping complex Dcp1/Dcp2 and decapping enhancers Edc3, Dhh1, and Edc2. (**A**) Smoothed, transcript-averaged PAR-CLIP occupancy profiles aligned at TSS and pA sites [±750 nt] of unstable and stable transcripts (first and fourth quantile of half-life distribution, respectively). (**B**) Dependence of total occupancy of factors on the transcripts half-life. The fitting function is plotted in red and the fitted value for b

*Figure 7 continued on next page*

*Figure 7 continued*

is marked with a dashed gray line. (Gray shade: 95% confidence intervals generated by bootstrapping transcripts).
(C) Sequence binding preference for the catalytically active subunit of decapping complex (Dcp2), illustrated with
the five most enriched and the 3 most depleted 4-mers. The color code shows the log2 enrichment factor of 4-
mers around PAR-CLIP cross-link sites [±8 nt]. Dark red represents strong enrichment and dark blue shows strong
depletion of a 4-mer. Infeasible combinations are shown with gray. The most highly enriched field is binding
AAAAU with the cross-link at the U, which is enriched over random expectation approximately $2^{2.3}$ = 5-fold.

DOI: https://doi.org/10.7554/eLife.47040.030

are consistent with a large body of published results and extend these to a global scale. In addition, the results revealed several unexpected, novel findings, which we discuss below. Although our data reflect factor cross-linking signal and measure occupancy on transcripts, and do not directly reveal function, the correlations of occupancies between factors and with transcript properties indicate functional aspects and suggest functional associations between factors that may guide future studies.

With respect to canonical mRNA turnover in the cytoplasm, the Pan2/Pan3 deadenylation complex and subunits of the Ccr4/Not complex bound preferentially to the pA site, reflecting a function in polyA tail shortening at early stages of RNA degradation. In contrast, the Ccr4/Not complex subunit Ccr4 bound also strongly in the 5′ region of transcripts. This pattern may reflect functional differences between Ccr4 and Pop2 during deadenylation (*Webster et al., 2018*) or an additional function of Ccr4 in transcription elongation (*Kruk et al., 2011*). Ccr4/Not subunits also show variations in their RNA-binding specificities, suggesting several isoforms of the complex that vary in composition and function, or RNA-specific conformational rearrangements. Factors involved in decapping show higher cross-linking near the RNA 3′ end, consistent with previously proposed binding near the pA site (*Chowdhury et al., 2007*). The profiles of decapping factors resemble those of Xrn1, suggesting formation of a complex with this 5′→3′ exonuclease or a fast mRNA decay by Xrn1 from 5′→3′ and slowing down towards the 3′ end. Complex formation of Xrn1 and the decapping factors is consistent with a function of Xrn1 in the buffering of mRNA levels in cells (*Sun et al., 2013*), which may be explained if Xrn1 is a regulatory component of the decapping complex.

The preferred localization of 5′ decapping and 3′ deadenylation factors near the mRNA 3′ and 5′ end, respectively, seems counterintuitive, but may be explained by formation of an mRNA closed-loop structure by messenger ribonucleoproteins (mRNPs), where 5′ and 3′ ends are in proximity (*Gallie, 1991*). It is possible that mature mRNPs carry decapping factors near their 3′ end, and that upon polyA tail shorting the decapping complex is activated, leading to decapping and rapid RNA degradation from the 5′ end. In this model, decapping would open the RNA closed-loop structure, providing access for exonucleases and allowing for rapid RNA removal. Further, our result that decapping factors are enriched on translationally non-optimal codons agrees with previous findings that suggest a link between translation and RNA degradation from the 5′ end through the decapping enhancer Dhh1 (*Radhakrishnan et al., 2016*). We note however that our approach does not detect binding events within the polyA tail, limiting further insights.

Comparison of our data with previous profiles of the nuclear surveillance factors Nrd1 and Nab3 reveals many factors that show a similar binding to aberrant non-coding nuclear RNAs, in particular antisense RNA upstream of known promoters, suggesting that these factors are part of the nuclear surveillance machinery. These results are consistent with published data (*Schmid and Jensen, 2018*) and with the following model for nuclear surveillance. First, the Nrd1/Nab3 complex recognizes aberrant, antisense RNAs (*Schulz et al., 2013*). These RNAs then get adenylated by the TRAMP4 complex, consistent with an interaction of the Nrd1/Nab3 complex and the TRAMP subunit Trf4 (*Tudek et al., 2014*). The short polyA tail targets the RNA for degradation by the nuclear exosome. We note that the degradation of introns and ncRNAs upstream of mRNAs on the same strand, which were annotated as NUTs and CUTs, is likely to use the same degradation mechanism because these are bound by the same factors (*Figure 2*, *Figure 3—figure supplement 2*). Thus, our results are consistent with the idea that degradation of short-lived ncRNAs in the nucleus involves Nrd1/Nab3, TRAMP4, and the exosome.

The exosome is involved in 3′→5′ cytoplasmic mRNA degradation, and in the processing and degradation of long-lived transcripts such as tRNAs, rRNAs and sn(o)RNAs. The differences in

exosome co-factor binding to different RNA classes (*Figure 2*) support the hypothesis that these factors confer specificity for processing and degradation of various RNA species (*Delan-Forino et al., 2017*). The exosome co-factor Ski2 shows cross-linking towards the 3′ end of mRNAs (*Figure 3*). This indicates that the subunit is important for initial RNA degradation by using its helicase activity to dissolve secondary structures and allowing the exosome to start degradation of the transcript (*Schneider and Tollervey, 2013*). Like the exosome complex, the exosome co-factor TRAMP5 binds mRNAs towards the 5′ end of transcripts, and thus some mRNAs may be targeted by TRAMP5 for exosomal degradation. The catalytic exosome subunits Rrp44 and Rrp6 and the exosome core cross-link near the mRNA 5′ end, probably because the exosome moves rapidly from 3′→5′ and then slows down, causing extensive cross-linking. In summary, our resource of transcriptome-binding profiles for 30 RNA degradation factors reveals several hypotheses for the function of these factors that can be tested case by case in the future.

# Materials and methods

**Key resources table**

| Reagent type (species) or resource | Designation | Source or reference | Identifiers | Additional information |
|---|---|---|---|---|
| Strain, strain background (*S. cerevisiae*, BY4741) | Ccr4_TAP | C-terminally tagged gene (Open Biosystems, Germany). | SGD: S000000019 | |
| Strain, strain background (*S. cerevisiae*, BY4741) | Pop2_TAP | C-terminally tagged gene (Open Biosystems, Germany). | SGD: S000005335 | |
| Strain, strain background (*S. cerevisiae*, BY4741) | Not1_TAP | C-terminally tagged gene (Open Biosystems, Germany). | SGD: S000000689 | |
| Strain, strain background (*S. cerevisiae*, BY4741) | Caf40_TAP | C-terminally tagged gene (Open Biosystems, Germany). | SGD: S000005232 | |
| Strain, strain background (*S. cerevisiae*, BY4741) | Pan2_TAP | C-terminally tagged gene (Open Biosystems, Germany). | SGD: S000003062 | |
| Strain, strain background (*S. cerevisiae*, BY4741) | Pan3_TAP | C-terminally tagged gene (Open Biosystems, Germany). | SGD: S000001508 | |
| Strain, strain background (*S. cerevisiae*, BY4741) | Dcp1_TAP | C-terminally tagged gene (Open Biosystems, Germany). | SGD: S000005509 | |
| Strain, strain background (*S. cerevisiae*, BY4741) | Dcp2_TAP | C-terminally tagged gene (Open Biosystems, Germany). | SGD: S000005062 | |
| Strain, strain background (*S. cerevisiae*, BY4741) | Edc2_TAP | C-terminally tagged gene (Open Biosystems, Germany). | SGD: S000000837 | |
| Strain, strain background (*S. cerevisiae*, BY4741) | Edc3_TAP | C-terminally tagged gene (Open Biosystems, Germany). | SGD: S000000741 | |
| Strain, strain background (*S. cerevisiae*, BY4741) | Dhh1_TAP | C-terminally tagged gene (Open Biosystems, Germany). | SGD: S000002319 | |
| Strain, strain background (*S. cerevisiae*, BY4741) | Xrn1_TAP | C-terminally tagged gene (Open Biosystems, Germany). | SGD: S000003141 | |
| Strain, strain background (*S. cerevisiae*, BY4741) | Rrp6_TAP | C-terminally tagged gene (Open Biosystems, Germany). | SGD: S000005527 | |

*Continued on next page*

*Continued*

| Reagent type (species) or resource | Designation | Source or reference | Identifiers | Additional information |
|---|---|---|---|---|
| Strain, strain background (*S. cerevisiae*, BY4741) | Csl4_TAP | C-terminally tagged gene (Open Biosystems, Germany). | SGD: S000005176 | |
| Strain, strain background (*S. cerevisiae*, BY4741) | Rrp40-TAP | C-terminally tagged gene (Open Biosystems, Germany). | SGD: S000005502 | |
| Strain, strain background (*S. cerevisiae*, BY4741) | Rrp4_TAP | C-terminally tagged gene (Open Biosystems, Germany). | SGD: S000001111 | |
| Strain, strain background (*S. cerevisiae*, BY4741) | Rrp44_TAP | C-terminally tagged gene (Open Biosystems, Germany). | SGD: S000005381 | |
| Strain, strain background (*S. cerevisiae*, BY4741) | Air1_TAP | C-terminally tagged gene (Open Biosystems, Germany). | SGD: S000001341 | |
| Strain, strain background (*S. cerevisiae*, BY4741) | Trf5_TAP | C-terminally tagged gene (Open Biosystems, Germany). | SGD: S000005243 | |
| Strain, strain background (*S. cerevisiae*, BY4741) | Mtr4_TAP | C-terminally tagged gene (Open Biosystems, Germany). | SGD: S000003586 | |
| Strain, strain background (*S. cerevisiae*, BY4741) | Air2_TAP | C-terminally tagged gene (Open Biosystems, Germany). | SGD: S000002334 | |
| Strain, strain background (*S. cerevisiae*, BY4741) | Trf4_TAP | C-terminally tagged gene (Open Biosystems, Germany). | SGD: S000005475 | |
| Strain, strain background (*S. cerevisiae*, BY4741) | Ski2_TAP | C-terminally tagged gene (Open Biosystems, Germany). | SGD: S000004390 | |
| Strain, strain background (*S. cerevisiae*, BY4741) | Ski3_TAP | C-terminally tagged gene (Open Biosystems, Germany). | SGD: S000006393 | |
| Strain, strain background (*S. cerevisiae*, BY4741) | Ski7_TAP | C-terminally tagged gene (Open Biosystems, Germany). | SGD: S000005602 | |
| Strain, strain background (*S. cerevisiae*, BY4741) | Ski8_TAP | C-terminally tagged gene (Open Biosystems, Germany). | SGD: S000003181 | |
| Strain, strain background (*S. cerevisiae*, BY4741) | Upf1_TAP | C-terminally tagged gene (Open Biosystems, Germany). | SGD: S000004685 | |
| Strain, strain background (*S. cerevisiae*, BY4741) | Upf2_TAP | C-terminally tagged gene (Open Biosystems, Germany). | SGD: S000001119 | |
| Strain, strain background (*S. cerevisiae*, BY4741) | Upf3_TAP | C-terminally tagged gene (Open Biosystems, Germany). | SGD: S000003304 | |
| Strain, strain background (*S. cerevisiae*, BY4741) | Nmd4_TAP | C-terminally tagged gene (Open Biosystems, Germany). | SGD: S000004355 | |
| Antibody | IgG | Sigma-Aldrich | Cat#: I5006, RRID:AB_1163659 | IP: 0.1 mg per IP |

*Continued on next page*

*Continued*

| Reagent type (species) or resource | Designation | Source or reference | Identifiers | Additional information |
|---|---|---|---|---|
| Antibody | PAP anti-TAP | Sigma Aldrich | Cat#: P1291, RRID:AB_1079562 | WB (1:2000) |
| Commercial assay or kit | Dynabeads Protein G | Invitrogen | Cat#: 10003D | 330 µl per IP |
| Commercial assay or kit | RNase T1 | Thermo Fisher Scientific | Cat#: EN0541 | |
| Commercial assay or kit | Antarctic P hosphatase | NEB | Cat#: M0289S | |
| Commercial assay or kit | RNase OUT | Invitrogen | Cat#: 10777019 | |
| Commercial assay or kit | T4 Polynucleotide Kinase | Invitrogen | Cat#: EK0032 | |
| Commercial assay or kit | T4 RNA ligase 2, truncated KQ | NEB | Cat#: M0373S | |
| Commercial assay or kit | T4 RNA ligase 1 | NEB | Cat#: M0437M | |
| Commercial assay or kit | Proteinase K | NEB | Cat#: P8107S | |
| Commercial assay or kit | SuperScript III RT | Thermo Fisher Scientific | Cat#: 18080093 | |
| Commercial assay or kit | Phusion High-Fidelity PCR Master Mix | Thermo Fisher Scientific | Cat#: F531S | |
| Chemical compound, drug | 4-thiouracil | Carbosynth | Cat#: 591-28-6 | 1 mM final conc. |
| Software, algorithm | mockinbird | *Roth and Torkler, 2018*; https://github.com/soedinglab/Degradation_scripts | | |
| Software, algorithm | UMI-tools | *Smith et al., 2017*; DOI: 10.1101/gr.209601.116 | | |
| Software, algorithm | Skewer | *Jiang et al., 2014*; DOI: 10.1186/1471-2105-15-182 | | |
| Software, algorithm | Bowtie | *Langmead et al., 2009*; DOI: 10.1186/gb-2009-10-3-r25 | | |
| Software, algorithm | tSNE | *Van Der Maaten and Hinton, 2008*; DOI: 10.1007/s10479-011-0841-3 | | |

## *S. cerevisiae* strain verification

*Saccharomyces cerevisiae* BY4741 strains harboring C-terminally tagged genes (Open Biosystems, Germany) were tested for the correctly inserted tag by Western Blotting using the Peroxidase Anti-Peroxidase (PAP; Sigma) antibody and Pierce ECL Western Blotting Substrate (Thermo Fisher Scientific, USA) (data not shown).

## PAR-CLIP experiments of *S. cerevisiae* proteins

PAR-CLIP was performed as described (*Baejen et al., 2014*; *Battaglia et al., 2017*). Briefly, TAP-tagged protein expressing yeast cells were grown in minimal medium (CSM mixture, Formedium, UK) supplemented with 89 µM uracil, 50–100 µM 4-thiouracil (4tU) and 2% glucose at 30℃ to $OD_{600}$ = 0.5. Cells were labeled in 1 mM 4tU final concentration for 4 hr. After labeling, cells were harvested, resuspended in ice-cold PBS and UV irradiated with 10–12 J/cm$^2$ at a wavelength of 365 nm on ice and continuous shaking. Lysis was performed in lysis buffer (50 mM Tris-HCl pH 7.5, 100 mM NaCl, 0.5% sodium deoxycholate, 0.1% SDS, 0.5% NP-40) by bead beating (FastPrep−24 Instrument, MP Biomedicals, LLC., France) using silica-zirconium beads (Roth, Germany). The cleared lysate was used for immunoprecipitation with rabbit IgG-conjugated Protein G magnetic beads

(Invitrogen, Germany) on a rotating wheel for 4 hr or overnight at 4°C. Beads were washed in wash buffer (50 mM Tris-HCl pH 7.5, 1 M NaCl, 0.5% sodium deoxycholate, 0.1% SDS, 0.5% NP-40). IP efficiency was controlled with part of the sample by Western Blot as shown in *Figure 1—figure supplement 2*. Partial digest of the cross-linked RNA was performed with 50 U RNase T1 per mL for 15–25 min at 25°C. The dephosphorylation reaction was performed in antarctic phosphatase reaction buffer (NEB, Germany) supplemented with 1 U/µL of antarctic phosphatase and 1 U/µL of RNase OUT (Invitrogen) at 37°C for 30 min. For rephosphyorylation, beads were incubated in T4 PNK reaction buffer A (Invitrogen) with a final concentration of 1 U/µL T4 PNK, 1 U/µL RNase OUT and 1 mM ATP for 1 hr at 37°C. 3′ adapter ligation was performed in T4 RNA ligase buffer (NEB) with 10 U/µL T4 RNA ligase 2 (KQ) (NEB), 10 µM 3′ adapter (5′ 5rApp-TGGAATTCTCGGGTGCCAAGG-3ddC 3′ (IDT), 1 U/µL RNase OUT, and 15% (w/v) PEG 8000 overnight at 16°C. 5′ adapter was ligated to the RNA using T4 RNA ligase buffer (NEB) with 6 U/µL T4 RNA ligase 1 (NEB), 10 µM 5′ adapter (5′ 5I nvddT-GUUCAGAGUUCUACAGUCCGACGAUCNNNNN 3′, IDT), 1 mM ATP, 1 U/µL RNase OUT, 5% (v/v) DMSO, and 10% (w/v) PEG 8000 for 4 hr at 25°C and 1 hr at 37°C. Beads were boiled in proteinase K buffer (50 mM Tris-HCl pH 7.5, 6.25 mM EDTA, 75 mM NaCl, 1% SDS) at 95°C for 5 min. Proteinase K digest was performed with 1.5 mg/mL proteinase K (NEB) for 2 hr at 55°C. RNA was recovered by acidic phenol/chloroform extraction followed by ethanol precipitation in presence of 0.5 µL GlycoBlue (Invitrogen) and 100 µM RT primer (5′ CCTTGGCACCCGAGAATTCCA 3′, IDT). SuperScript III RTase was used for reverse transcription for 1 hr at 44°C and 1 hr at 55°C. NEXTflex barcode primer and universal primer were added to cDNA by PCR amplification with Phusion HF master mix (NEB). After PCR amplification, cDNA was purified and size-selected on a 4% E-Gel EX Agarose Gel (Invitrogen). Quantification on an Agilent 2200 TapeStation instrument (Agilent Technologies, Germany) and 50–75 nt single-end sequencing was performed on Illumina sequencers (HiSeq1500, HiSeq2500 and NextSeq550).

## PAR-CLIP data pre-processing

Reads from PAR-CLIP experiments with replicates were merged after making sure that all samples showed high Spearman correlation values comparing binding occupancies of replicates on different genes (*Figure 2—figure supplement 2*). Mapping and statistical evaluation of PAR-CLIP experiments was performed using our in-house software mockinbird (*Roth and Torkler, 2018*). In summary, the UMI is removed from the 5′ end with UMI-tools (*Smith et al., 2017*), and the 3′ adapter is trimmed with Skewer (*Jiang et al., 2014*). Reads with traces of the 5′ adapter are discarded. The preprocessed reads are then mapped to the *S. cerevisiae* genome (sacCer3, version 64.2.1). After mapping PCR duplicates are removed with UMI-tools.

We used two alternative approaches for mapping reads using Bowtie (*Langmead et al., 2009*): For all analyses except the 'transcript class enrichment analysis' in *Figure 2*, reads are uniquely mapped with up to one mismatch. We discard alignments shorter than 20 nt. This stringent mapping ensures that our high confidence PAR-CLIP cross-link sites are originating from correctly mapped reads on the reference genome. For *Figure 2*, unique mapping would cause the loss of most reads that fall into rRNAs and tRNAs because of duplicated rRNA genes and tRNA isodecoders. For *Figure 2*, we therefore allowed Bowtie multi-mapping in two regions with –best, –starra options and discarded reads shorter than 30 nt.

T→C transitions directly at the edge of the reads or with a Phred quality score lower than 20 are not considered as signature of protein binding as they suffer from higher technical noise. To obtain high confidence cross-link sites, we set a stringent cutoff of 0.005 for the p-value of cross-link sites and require a minimum coverage of 2 per site. Moreover, if we see the same transition in at least 75% of reads in the input library control (SRA: SRX532381) (*Baejen et al., 2014*), we annotate it as a single nucleotide polymorphism of our lab strain with respect to the genomic reference and remove such sites from our analysis. Finally, the occupancy of a factor on a verified cross-link site is defined as the number of transitions obtained from our PAR-CLIP experiments divided by the concentration of RNAs covering the cross-link site according to the input library control. This control coverage is measured under comparable conditions to PAR-CLIP experiments (*Baejen et al., 2014*). Occupancy values are capped at the 95th percentile. Subsequent analyses were performed using in-house python scripts. Mockinbird configuration files as well as the analysis scripts can be found at https://github.com/soedinglab/Degradation_scripts (copy archived at https://github.com/elifesciences-publications/Degradation_scripts).

## Transcript class enrichment

We analyzed the distribution of reads from high confidence cross-link sites over the genome (*Figure 2A*). We presented the sum of reads from 5′ and 3′ UTRs, coding sequences, and introns as the value for mRNAs. Reads that fall within genomic regions not annotated as categories analyzed here are shown with gray. These annotated transcript classes have comparable U-content, making the comparison between fractions of cross-link sites in each category possible (*Figure 2—figure supplement 2*).

For each factor studied here, we defined enrichment scores that represent their preferences for binding to various transcript classes *c*, in comparison to all other factors. We use annotations for rRNA, tRNA, snoRNA, snRNA, coding sequences (CDS), from *S. cerevisiae* genome sacCer3, version 64.2.1. Untranslated regions around coding boundaries (5′ and 3′ UTRs) were annotated based on TIF-seq experiment (*Pelechano et al., 2013*). We selected the most strongly expressed isoform for each gene. We then assigned boundaries to 3′ and 5′ UTRs based on annotated CDS of the same gene. We furthermore used annotations for stable, unannotated transcripts (SUTs), cryptic unstable transcripts (CUTs), and Nrd1-unterminated transcripts (NUTs) (*Neil et al., 2009*; *Pelechano et al., 2013*; *Schulz et al., 2013*). We removed overlapping annotations with the following priority list: rRNA, tRNA, snRNA, snoRNA, intron, CDS, UTR, SUT, CUT, NUT. For each factor, we counted the number of high confidence reads falling in each transcript class. We then used the $\log_2$-transformed matrix and normalized it in the following way for both rows and columns to get log enrichment values that sum to zero in both rows and columns. The row- and sum-normalized enrichment score is defined as follows, where $X_{f,c}$ is the number of high confidence reads for factor *f* that fall into transcript class *c*, and $X'_{f,c} = \log_2 X_{f,c}$ (*Figure 2B*):

$$\tilde{X_{f,c}} = X'_{f,c} - \frac{X'_{f,\circ} X'_{\circ,c}}{X'_{\circ,\circ}}$$

We defined the row and sum averages of $X_{f,c}$,

$$X'_{f,\circ} = 1C \sum_{c=1}^{C} X'_{f,c},$$

$$X'_{\circ,c} = 1F \sum_{f=1}^{F} X'_{f,c},$$

$$X'_{\circ,\circ} = 1FC \sum_{f=1}^{F} \sum_{c=1}^{C} X'_{f,c},$$

F is the number of factors and C is the number of transcript classes (*Figure 2B*). The normalization can be interpreted as subtracting from the log enrichment matrix X′ the first singular component of its singular-value decomposition.

## Metagene analysis

We used the most abundant TIF-annotated isoform for mRNAs (*Pelechano et al., 2013*) as a reference. Transcripts longer than 1500 bases are chosen and aligned at their TSS or pA sites. The average occupancy per nucleotide is then calculated based on high confidence cross-link sites of each PAR-CLIP experiment. The profiles are smoothed by a moving average in a 41 nt window and the 95% confidence interval is estimated by 1500 bootstrap sampling iterations over the transcripts. To further denoise the profiles, the cross-link sites falling in snRNAs, rRNAs, and tRNAs are removed. Furthermore, to avoid ambiguous results, we made sure that the profile comes solely from the central gene. To do so, we performed the metagene analysis around the TSS on the sense strand on TIF-annotated mRNAs that have no other mRNA up to 700 bp upstream of their TSS (3193 transcripts in total). Analogously, for sense-strand pA site profiles we used mRNAs that have no nearby genes downstream of their pA site up to 700 bases on the same strand (3193 transcripts in total). For the antisense strand profiles, we applied the same criteria on the opposite strand which left us

with 3076 and 3193 transcripts filtered around TSS and pA site respectively. This ensures that the observed antisense binding does not originate from neighboring or overlapping transcripts on the antisense strand. In both cases we looked at the average occupancy in a window of [±700 nt] around TSS and around pA sites. Occupancies were normalized to the maximum value, which is the background binding level for antisense profiles with no significant cross-linking to the antisense strand (*Figure 3*, *Figure 4*, *Figure 3—figure supplement 3*, and *Figure 4—figure supplement 2*). The same procedure was followed to plot metagene occupancies centered around protein-coding regions and snoRNAs from *S. cerevisiae* genome sacCer3, version 64.2.1 (*Figure 3—figure supplement 1* and *Figure 2—figure supplement 1*). Similarly, CRAC coverage profiles of Xrn1, Mtr4, Trf4, and Ski2 (pre-processed as described in *Tuck and Tollervey, 2013*) were aligned to TIF-annotated transcripts in the same approach as described here (*Figure 1—figure supplement 1*).

## Co-occupancy

Co-occupancy measures the tendency of two factors to bind to the same transcripts. Occupancy of a factor on a transcript is defined as the sum of occupancies for all high confidence cross-link sites falling within this transcript. Co-occupancy of two factors is defined as the Pearson correlation over all transcripts between the occupancies of these factors (*Figure 5A*). We used these correlation values between all pairs of RNA processing factors to assign distances to each pair and used tSNE (*Van Der Maaten and Hinton, 2008*) to visualize the two-dimensional nonlinear embedding of co-occupancies for all RNA-binding proteins in our dataset (*Figure 5C*).

## Co-localization

Co-localization measures how likely two factors are to bind near each other in the transcriptome. More precisely, we first calculate the occupancy of a factor $f \in \{1,\ldots,F\}$ around the cross-link sites of another factor $f'$ ([−40 nt,+40 nt] excluding the centered T). We then normalize according to the total occupancy values,

$$z_{ff'} = \sum_{i=1}^{n_{f'}} \left( \sum_{j=-40}^{-1} Occ_{ff',i,j} + \sum_{j=1}^{40} Occ_{ff',i,j} \right)$$

$$\text{co} - \text{localization}(f,f') = \frac{z_{ff'}}{\sum_f z_{ff'} \sum_{f'} z_{ff'}}$$

Where, $n_f$ is the number of cross-link sites for factor $f$, and $Occ_{ff',i,j}$ is the occupancy of $f$ at position $j$ around the $i^{th}$ cross-link site from factor $f'$ ($Occ_{ff',i,j} = 0$ if no verified cross-link sites exist). To improve signal-to-noise, we compute from the resulting matrix of co-localizations between all RNA-processing factors $C_{f,f'}$, the matrix of Pearson correlations between the rows of $C_{f,f'}$, (*Figure 5B*).

## Codon-enrichment analysis

To search for possible links between translation efficiency and RNA degradation, we checked if some degradation factors preferentially bind to translationally efficient/non-efficient transcripts. To do so we adapted the proposed normalized translation efficiency scale (*Pechmann and Frydman, 2013*). The authors generate a normalized optimality score for codons that incorporates the competition between supply and demand of tRNAs. The coding region for each transcript was extracted according to ORFs annotated by SGD. The codon optimality score was averaged over the whole reading frame (*Figure 6A*, more detailed explanation in the next section).

We then checked whether mRNAs that bind to each factor are enriched or depleted in some codons compared to all mRNAs. To achieve this, we defined the following score for codon enrichment that represents deviations from average frequencies in all mRNAs,

$$codon\,enrichment = \frac{\sum_{t=1}^{T} \left( \frac{occ(t)}{\sum_{t'=1}^{T} occ(t')} \times F_{c,t} \right)}{\frac{1}{T} \sum_{t=1}^{T} F_{c,t}}$$

Here $T$ is the number of mRNA transcripts, $F_{c,t}$ is the fraction of the codon $c$ in transcript $t$, and $occ(t)$ is the *total* occupancy of the factor on transcript $t$. 90% confidence intervals were

generated by bootstrapping: we sampled *with replacement* 1000 times the same number of mRNAs from the total set as in total, and for each set we recalculated the codon enrichment score. We colored the bars based on the previously ranked optimality of codons (*Pechmann and Frydman, 2013*) (*Figure 6A*, *Figure 6—figure supplements 1–7*).

## Relating occupancies to various transcript features

We analyzed the correlation of the occupancy of all factors with transcript length, codon enrichment of the transcript, expression level, transcript stability, and polyA tail length. For expression, we used an RNA-seq experiment of wild-type yeast (SRA: SRX532381) (*Baejen et al., 2017*) and mapped the reads to mRNAs. We present the average number of reads per base as an estimate for gene expression. For half-life calculations, we used published yeast 4tU-seq (GEO: GSM2199309) and RNA-seq experiments (SRA: SRX532381) (*Baejen et al., 2017*). Transcript half-life is estimated with an optimized method that will be published elsewhere (Hofmann et al., unpublished).

Since there are only few transcripts with very low or very high half-life, codon optimality, and expression (*Figure 6—figure supplement 8*), we performed the analysis on a subset of mRNAs where the transcript property lies between the 5% and 95% quantiles. We then compared the *total* occupancy of degradation factors on each mRNA relative to such transcript features (*Figures 6A* and *7B*, and *Figure 6—figure supplements 1–7*). We show 95% confidence intervals generated by bootstrapping mRNAs in gray shade.

We checked whether such correlations originate from the feature of interest or merely shows up due to correlations between this feature and others (*Figure 6—figure supplement 8*). We used a multivariate linear regression to model total occupancy as a linear function of these four features:

$$occupancy'(t) \sim length + optimality + expression + half\,life$$

In cases where the correlation is a direct effect from our feature of interest, we expect to lose significantly on our prediction when this variable is taken out of the equation. Therefore, we use p-values representing the importance of each feature in this linear regression as a score representing the significance of its contribution in explaining the final occupancies. Occupancy correlated strongly with transcript length, which dominated as explanatory variable in this regression, trivially because most factors bind along the entire transcript. To eliminate this trivial dependency, we used occupancy per nucleotide, denoted $occupancy'$, as the target variable in our regression (*Figure 6C*).

## Motif enrichment analysis

To find sequence preferences for binding events of degradation factors, we counted 4-mers in a window of [±5 nt] intervals around high confidence cross-link sites of PAR-CLIP experiments. Based on this count table, the enrichment score for each 4-mer was calculated using the following formula,

$$enrichment\,(4-\mathrm{mer}, i) = \frac{n_{4-\mathrm{mer},i} + 1}{N \times \Pi_{j=1}^{4} P_{4-\mathrm{mer}[j]}}$$

Here $N$ is the number of cross-link sites below the cut-off p-value (we used a maximum of 5000 cross link sites), $n_{4mer,\,i}$ is the number of observed *4-mers* at position $i$ in the set of binding sequences aligned at their cross-link site $i$=0, 4-*mer*[j] is the base at the $j$'th position of the 4-mer, and $P_b$ is the probability of observing base $b$. We used the probabilities: $P_A = P_T = 0.31$ and $P_C = P_G = 0.19$ based on frequencies in yeast genome and corrected for the T bias at the cross-link site (*Figure 4—figure supplement 1*).

## Acknowledgements

We would like to thank Helmut Blum (LAFUGA, LMU Munich), Stefan Krebs (LAFUGA, LMU Munich), Kerstin Maier and Petra Rus (Cramer laboratory) for sequencing. We thank Gabriel Villamil and Bjoern Schwalb (Cramer laboratory) for sharing RNA half-life calculations prior to publication. PC was funded by the Advanced Grant TRANSREGULON of the European Research Council and by the Volkswagen Foundation. This work was supported by the DFG SPP1935 grant CR 117/6–1.

## Additional information

### Funding

| Funder | Grant reference number | Author |
|---|---|---|
| European Research Council | Advanced Grant Transregulon | Patrick Cramer |
| Volkswagen Foundation | | Patrick Cramer |
| Deutsche Forschungsge-meinschaft | SPP1935 grant CR 117/6-1 | Johannes Soeding Patrick Cramer |
| Max-Planck-Gesellschaft | Open-Access funding | Patrick Cramer |

The funders had no role in study design, data collection and interpretation, or the decision to submit the work for publication.

### Author contributions

Salma Sohrabi-Jahromi, Formal analysis, Validation, Investigation, Visualization, Methodology, Writing—original draft, Writing—review and editing; Katharina B Hofmann, Data curation, Validation, Investigation, Methodology, Writing—original draft, Writing—review and editing; Andrea Boltendahl, Saskia Gressel, Carlo Baejen, Data curation, Methodology; Christian Roth, Formal analysis, Visualization, Methodology; Johannes Soeding, Conceptualization, Supervision, Funding acquisition, Methodology, Writing—original draft, Project administration, Writing—review and editing; Patrick Cramer, Conceptualization, Supervision, Funding acquisition, Writing—original draft, Project administration, Writing—review and editing

### Author ORCIDs

Salma Sohrabi-Jahromi (iD) https://orcid.org/0000-0002-8417-8230
Katharina B Hofmann (iD) https://orcid.org/0000-0002-0683-6277
Saskia Gressel (iD) http://orcid.org/0000-0003-0261-675X
Patrick Cramer (iD) https://orcid.org/0000-0001-5454-7755

### Decision letter and Author response

Decision letter https://doi.org/10.7554/eLife.47040.044
Author response https://doi.org/10.7554/eLife.47040.045

## Additional files

### Supplementary files

• Supplementary file 1. Overview of RNA processing factors and their respective PAR-CLIP experiments used in this study.
DOI: https://doi.org/10.7554/eLife.47040.031

• Transparent reporting form
DOI: https://doi.org/10.7554/eLife.47040.032

### Data availability

Sequencing data have been deposited in GEO under accession codes GSE 128312.

The following dataset was generated:

| Author(s) | Year | Dataset title | Dataset URL | Database and Identifier |
|---|---|---|---|---|
| Sohrabi-Johromi S, Hofmann KB, Boltendahl A, Roth C, Gressel S, Baejen C, Soeding J, Cramer P | 2019 | Transcriptome maps of general eukaryotic RNA degradation factors | https://www.ncbi.nlm.nih.gov/geo/query/acc.cgi?acc=GSE128312 | NCBI Gene Expression Omnibus, GSE8128312 |

The following previously published datasets were used:

| Author(s) | Year | Dataset title | Dataset URL | Database and Identifier |
|---|---|---|---|---|
| Schulz D, Schwalb B, Kiesel A, Baejen C, Torkler P, Gagneur J, Soeding J, Cramer P | 2013 | Nuclear depletion of the essential transcription termination factor Nrd1 in Saccharomyces cerevisiae was studied using a combination of RNA-Seq, ChIP-Seq of Pol II and PAR-CLIP of Nrd1 | https://www.ebi.ac.uk/arrayexpress/experiments/E-MTAB-1766/ | Array Express, E-MTAB-1766 |
| Baejen C, Torkler P, Gressel S, Essig K, Söding J, Cramer P | 2014 | Transcriptome maps of mRNP biogenesis factors define pre-mRNA recognition | https://www.ncbi.nlm.nih.gov/geo/query/acc.cgi?acc=GSE59676 | NCBI Gene Expression Omnibus, GSE59676 |
| Baejen C, Andreani J, Torkler P, Battaglia S, Schwalb B, Lidschreiber M, Maier KC, Boltendahl A, Rus P, Esslinger S, Soeding J, Cramer P | 2017 | Genome-wide analysis of RNA polymerase II termination at protein-coding genes. | https://www.ncbi.nlm.nih.gov/geo/query/acc.cgi?acc=GSE79222 | NCBI Gene Expression Omnibus, GSE79222 |
| Battaglia S, Lidschreiber M, Baejen C, Torkler P, Vos S, Cramer P | 2017 | RNA-dependent chromatin association of transcription elongation factors and Pol II CTD kinases | https://www.ncbi.nlm.nih.gov/geo/query/acc.cgi?acc=GSE81822 | NCBI Gene Expression Omnibus, GSE81822 |

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
