## [Decision Letter]

[Editors’ note: a previous version of this study was rejected after peer review, but the authors submitted for reconsideration. The first decision letter after peer review is shown below.]

Thank you for submitting your work entitled "Transcriptome maps of general eukaryotic RNA degradation factors" for consideration by *eLife*. Your article has been reviewed by three peer reviewers, one of whom is a member of our Board of Reviewing Editors, and the evaluation has been overseen by a Senior Editor. The following individual involved in review of your submission has agreed to reveal their identity: David Tollervey (Reviewer #2).

Our decision has been reached after consultation between the reviewers. All reviewers acknowledge the extensive data sets obtained, the quality of the analysis and the potential value for the RNA turnover field. There are, however, concerns about the reproducibility of the data and the significance of the biological insights, given the absence of any independent validation of the results. We are therefore obliged to reject this paper for publication in *eLife*. However, new submission of a revised manuscript would be welcome if all the reviewers' concerns can be addressed.

We hope that the individual reviews appended below will be helpful in deciding how to move forward.

*Reviewer #1:*

In this manuscript the authors describe the genome-wide binding sites for 30 different proteins acting in multiple RNA degradation pathways. The data will be a valuable resource for researchers in this area and should be published. However, there are a numerous places were the manuscript needs to be improved by relatively minor rewrites such that the text more accurately reflects the data. Some of the issues are:

1) The description of the reliability and quality of the data in the "Transcriptome maps for 30 RNA degradation factors" section of the Results is not completely congruent with the results.

1a) "The high reproducibility of these PAR-CLIP experiments is revealed by a comparison of biological replicates that we collected for eleven out of the 30 degradation factors (Figure 1—figure supplement 1)." The supplementary figure actually shows high reproducibility for 4 of the proteins (all cytoplasmic decapping factors), medium reproducibility for 3 factors and modest reproducibility for 4 factors.

1b) 19 of the factors were not analyzed in duplicate and thus the reproducibility of these datasets is hard to evaluate.

1c) Furthermore, the Materials and methods indicate that "Reads from PAR-CLIP experiments with replicates were merged", thus it is not clear what part of the analyses were actually preformed in duplicate.

2) At least seven of the factors studied here have been previously studied by the Tollervey lab (in what they call CRAC, which is fundamentally the same as PAR-CLIP). Thus, a valuable extension of this study and the validity of the data would be to compare reproducibility between the author's PAR CLIP data and the published CRAC data.

3) "we typically obtained tens of thousands of verified factor-RNA cross-link sites (Table 1)." Table 1 actually shows that about half of the factors meet this tens of thousands threshold while the remaining factors vary between 561 and 16663 sites.

4) In the description of "RNA end-processing complexes differ in their targets" section the highlighted results that cytoplasmic factors like Caf40 and Ccr4 bind sparsely to nuclear RNAs such as introns, snRNAs and snoRNAs is not surprising. The authors attribute this to a "strong binding preference", but this may simply reflect that these factors are not exposed to these RNAs instead of binding constants.

More importantly, the z-scoring in Figure 1B appears to be inappropriate for these kinds of statements. If I understand the description of the z-scoring correctly the unexpected high z-score of Ccr4 for tRNAs in Figure 1B means that tRNAs have a strong preference for Ccr4 compared to the other 29 factors, not that Ccr4 has a strong preference for tRNAs compared to all other RNAs as implied in the Results section. Another example of problematic description of Figure 1B is "Rrp6.…. shows binding to rRNAs, tRNAs, snoRNAs and snRNAs." However Figure 1A does not appear to show appreciable binding to rRNAs.

5) "Dcp2, which harbors the hydrolase activity that removes the cap, cross-links frequently to mRNAs, in particular their 3 ´UTR, and to short noncoding transcripts, in especially to unstable SUTs. This suggests that Dcp2 is involved not only in cytosolic mRNA degradation but also in decapping during nuclear RNA surveillance." However, the defining characteristic that differentiates SUTs from CUTs is that SUTs are degraded by cytoplasmic Dcp2-dependent pathways as shown by several different labs (e.g. Pubmed IDs 17074811, 21610164, 21826286, 24931603). Thus rather than suggesting a new role for Dcp2, these findings confirm a different known role.

6) "The analysis recovers established interactions between subunits of known complexes" but this is not supported by any quantitative analysis. Among the factors the authors analyze, some proteins that are thought to form stable complexes indeed cluster (e.g. Dcp1/2, Pan2/3) while other known stable complexes do not (e.g. Ski2/3/8 or the Air1 subunit of TRAMP).

7) Some of the names of the clusters in Figure 4C are poorly justified. For example, out of the six proteins in the "nuclear ncRNA processing" cluster four are cytoplasmic proteins that act mostly or exclusively on mRNA (Ccr4, Upf1, Ski3 and Ski7). They also function in degradation, rather than processing, thus at least the first two but possibly all three words in this name should be changed.

8) "For decapping factors (Dcp1, Edc3, Edc2, and Dhh1) and Xrn1, low codon optimality is the most determining feature for binding (Figure 5C).” However Figure 5C appears to show that this is true for three of these factors, but the opposite is true for Edc2, where high codon optimality appears to be determining. The authors should clarify.

9) "Moreover, A-rich 4-mers are abundant around the proximity ( ± 8 nt) of Dcp2-cross-link sites (Figure 6C)." However Figure 6C and the legend say the window is ± 5 instead of ± 8. This might be a typo.

10) "As expected, most PAR-CLIP cross-link sites fall into the mRNA transcript class, although about one third of the factors also show a considerable number of cross-links to non-coding RNAs". I suggest deleting "as expected" or replacing it with "surprisingly". Several of the factors have known roles in ncRNA processing/degradation and mRNA is only a minor fraction of cellular RNA and therefore I would not have expected most sites to fall into the mRNA class.

A related issue may be that the analysis in Figure 1 appears to only include "Fractions of high-confidence PAR-CLIP cross-link sites" and not the actual number of or fraction of high-confidence reads. e.g. if there are 9 sites, in mRNA with 100 reads each and one in a ncRNA with 900 reads this would count as 90% of the sites being in mRNA, but only 50% of the reads are in mRNA. Since the complexity of mRNA is probably the highest (more than half the genome), the number of distinct sites in mRNA for a given factor is probably a systematic overrepresentation of that factor's importance in mRNA processing.

11) In Materials and methods "various non-overlapping transcript classes c" needs to be clarified. Several of the transcript classes in fact do overlap in the genome. Many snoRNAs are encoded within introns. Many CUTs, SUTs and NUTs overlap with mRNAs and each other. The methods need to be clarified.

*Reviewer #2:*

The authors present the PAR-CLIP analysis of 30 proteins implicated in RNA turnover or surveillance. In principal, these results should be of considerable value to the field, but this is much reduced by the lack of biological or even technical replicates for most of the experiments. Moreover, there are some concerns about the mapping of the data to the yeast genome. The downstream bioinformatic analysis of the data appears to have been very well performed, with imaginative approaches being applied and interesting conclusions. The 2D clustering analysis (Figure 4C and Figure 4—figure supplement 3) is particularly good.

I regret that the technical problems with the manuscript prevent me from recommending acceptance of the current version.

Specific points:

1) Subsection “Transcriptome maps for 30 RNA degradation factors”: It appears that biological replicates were performed for only eleven out of the 30 degradation factors tested. For 19 of the factors analyzed, the conclusions are seemingly based on single experiments. While it is the case that the experiments reported involve a lot of work, it is quite unusual for the bulk of data in a manuscript to be published without clearer evidence of reproducibility. It would have been better to analyze fewer proteins in more depth. The authors present data indicating the consistency of the results from the proteins for which replicates were performed. However, while the replicates appear to cluster well in the 2D diagrams, inspection of the metagene analysis reveals appreciable differences between duplicates (Figure 1—figure supplement 1), which would presumably be much more marked at the level of individual genes.

This is particularly an issue when the results are not as expected, e.g. the apparent binding of the deadenylase Ccr4 to a high proportion of tRNAs, which is based only a total of only 4,540 confident Ccr4 cross-linking sites.

2) The authors report that "most PAR-CLIP cross-link sites fall into the mRNA transcript class". This is unexpected, particularly for pre-rRNA processing factors, including the exosome and Mtr4, that should recover high level of rRNA hits. Inspection of the methods indicates that the reads were mapped to a unique database. This would have the effect of eliminating most rRNA and tRNA reads, as well as a subset of mRNAs. In the yeast genomic database, there are 2 copies of the rDNA repeat, so unique mapping will eliminate most rRNA hits. For tRNAs the situation is more complex; most tRNAs are present in multigene families and will be eliminated from unique hits, whereas the few single copy tRNA genes will be retained. This might be a particular issue for the reported Ccr4-tRNA interactions. For mRNAs, duplicated genes such as R-proteins will also be excluded.

This point should be clarified. If uniquely-mapped reads were indeed used, the full mapping data should be presented.

3) Subsection “RNA end-processing complexes differ in their targets”: SUTs can be cytosolic, so this might not be sufficient to show that Dcp2 has nuclear activity.

4) Subsection “Distinct factor distribution along mRNA”, Figure 1B (and end of subsection “PAR-CLIP experiments of *S. cerevisiae* proteins”): It would be helpful to discuss more why Csl4, Rrp4 and Rrp40 have distinct profiles. In particular, the Rrp4 profiles looks very different from all other degradation factors. This raises question of the PAR-CLIP experiment really worked for each cofactor. It is notable that some of these have low numbers of XL sites (561 for Rrp6, 923 for Rrp40, 7,373 for Csl4, while Rrp44 has much more 44,721).

5) Subsection “Interactions between RNA processing machineries”, Figure 4 and Figure 4—figure supplements 1-2: It would be helpful to discuss why Trf4/Air2 is bound to introns (Figure 1) but not Trf5 while Trf5/Air1 is showing higher co-occupancy / colocalization with splicing factors

*Reviewer #3:*

The paper by Sohrabi-Jahromi et al. presents genome-wide RNA binding datasets for *S. cerevisiae* RNA decay factors. Analyses of the obtained data are often in-line with current knowledge but also uncover several unexpected binding patterns.

The presented data are certainly of interest to a wide audience. However, there is no independent validation of any of the unexpected binding patterns. This is especially problematic, given that many datasets are based on single biological replicates. Moreover, some observations challenge our current thinking about the function of the investigated factor(s), which, to be suitable for publication in *eLife*, should be supported by independent experiments.

- Functional relevance:

The manuscript presents a series of unexpected and somewhat controversial observations. One example concerns a possible role for Pop2 and Not1 in decay of CUTs. Any functional relevance of such binding should be trivial to validate. This would be relevant for other of the most striking findings. Such validation may be possible by re-analysis of published datasets, for example by intersecting occupancy with RNAseq upon depletion/mutation of the same factors.

- Replicates and Technical artefacts:

PAR-CLIP is a complex technique, where technical artefacts are not uncommon. For example, binding to highly abundant tRNA, snRNA and snoRNA species is often observed, even for factors where such binding is unlikely to occur or be relevant in vivo. Hence, it is quite possible that binding to such species, especially when based on a single biological replicate, does not reflect biology. Such considerations should be mentioned in the manuscript. Ideally, all factors for which single replicates are presented, should be backed up by additional data. This may be simply an additional replicate, or some kind of independent validation.

More generally, more information on how the authors scrutinize the quality of their PAR-CLIP experiments should be given (i.e. more explicitly than simply referring to another publication).

[Editors’ note: what now follows is the decision letter after the authors submitted for further consideration.]

Thank you for resubmitting your work entitled "Transcriptome maps of general eukaryotic RNA degradation factors" for further consideration at *eLife*. Your revised article has been favorably evaluated by James Manley as the Senior Editor, and three reviewers, one of whom is a member of our Board of Reviewing Editors.

The manuscript has been considerably improved but there are some remaining issues that need to be addressed before acceptance, as outlined below:

1) Even though some of the more controversial suggestions of the original draft are now resolved, comparison with available knock-out or mutant / RNA-seq datasets would greatly help to support, or dismiss, some of the mentioned novel findings.

2) If available, western blots and/or autoradiographs of the different PAR-CLIP experiments would be nice to include to allow readers to judge the quality of different IP experiments.

3) Some of the conclusions presented in the Results sections still appear vague or unclear. This concerns for example interpretation of the RNA exosome binding profiles.

a) i.e. In the first paragraph of the subsection “The exosome and surveillance factors”: unclear what different binding patterns "reflect" the distinct functions of the RNA exosome.

b) In the second paragraph of the subsection “Distinct factor distribution along mRNA”: How/why would TRAMP4 binding upstream of the TSS affect early termination of the mRNA? Isn't it more plausible that it serves a role in termination of (unannotated) upstream ncRNA? Similarly, 5' bias of exosome and TRAMP subunits could arise from unannotated NUT-like transcripts.

4) Figure 3 and Figure 3—figure supplement 1B: What RNAs generate the high signals for Mtr4 and Air2 5' to the TSS? This region is expected to give rise to very few transcripts. The upstream ncRNAs are further upstream and on the opposite strand. There must be a concern that some sequences have been mis-mapped. Were the sequences mapped only to a small number of loci, which would strengthen this concern? The authors should re-check the data supporting these distributions.

5) The second paragraph of the subsection “The exosome and surveillance factors” refers to Figure 2—figure supplement 3 and not Figure 3—figure supplement 3.

6) Figure 5C: 3' degaradation => 3' degradation

---

## [Author Response]

[Editors’ note: the author responses to the first round of peer review follow.]

Reviewer #1:1) The description of the reliability and quality of the data in the "Transcriptome maps for 30 RNA degradation factors" section of the Results is not completely congruent with the results.1a) "The high reproducibility of these PAR-CLIP experiments is revealed by a comparison of biological replicates that we collected for eleven out of the 30 degradation factors (Figure 1—figure supplement 1)." The supplementary figure actually shows high reproducibility for 4 of the proteins (all cytoplasmic decapping factors), medium reproducibility for 3 factors and modest reproducibility for 4 factors.1b) 19 of the factors were not analyzed in duplicate and thus the reproducibility of these datasets is hard to evaluate.1c) Furthermore, the Materials and methods indicate that "Reads from PAR-CLIP experiments with replicates were merged", thus it is not clear what part of the analyses were actually preformed in duplicate.

Thank you for this important comment, which we took very seriously. To ensure the highest possible data quality and reproducibility, we generated 49 new datasets, obtaining between two and four highly reproducible biological replicates for each degradation factor (Figure 1—figure supplement 1). Upon comparing the replicates, we found poor reproducibility for Pop2, Not1, Rrp4 and Caf40 and therefore omitted old replicates from the analysis. Furthermore, we applied a more conservative threshold for including data and eliminated six of the previous replicates that had less than 10.000 cross-link sites (Csl4, Rrp40, Rrp6, Ski2, Ski3, and Ski7). As a result, only highly reproducible data with a high number of crosslink sites were included in the final data set. This has led to a clearly visible improvement of the analysis and revised manuscript.

We now write:

“The high reproducibility of these PAR-CLIP experiments is revealed by a comparison of two independent biological replicates that we collected for all 30 degradation factors (Figure 1—figure supplement 1), with Spearman correlations between 0.87 and 1.00 (mean: 0.94).”

The extent of improvement of the PAR-CLIP data can be appreciated by comparing the old Figure 2 with new Figure 3.

2) At least seven of the factors studied here have been previously studied by the Tollervey lab (in what they call CRAC, which is fundamentally the same as PAR-CLIP). Thus, a valuable extension of this study and the validity of the data would be to compare reproducibility between the author's PAR CLIP data and the published CRAC data.

We have included the comparison between metagene profiles obtained from CRAC data and PAR-CLIP in Figure 1—figure supplement 1. The agreement between these studies is indeed good.

3) "we typically obtained tens of thousands of verified factor-RNA cross-link sites (Table 1)." Table 1 actually shows that about half of the factors meet this tens of thousands threshold while the remaining factors vary between 561 and 16663 sites.

Compare comment above. We have generated new biological replicates for most of the PAR-CLIP experiments. Our new datasets contain between 21,311 and 590,328 cross-link sites per factor (Figure 1).

4) In the description of "RNA end-processing complexes differ in their targets" section the highlighted results that cytoplasmic factors like Caf40 and Ccr4 bind sparsely to nuclear RNAs such as introns, snRNAs and snoRNAs is not surprising. The authors attribute this to a "strong binding preference", but this may simply reflect that these factors are not exposed to these RNAs instead of binding constants.More importantly, the z-scoring in Figure 1B appears to be inappropriate for these kinds of statements. If I understand the description of the z-scoring correctly the unexpected high z-score of Ccr4 for tRNAs in Figure 1B means that tRNAs have a strong preference for Ccr4 compared to the other 29 factors, not that Ccr4 has a strong preference for tRNAs compared to all other RNAs as implied in the Results section. Another example of problematic description of Figure 1B is "Rrp6.…. shows binding to rRNAs, tRNAs, snoRNAs and snRNAs." However Figure 1A does not appear to show appreciable binding to rRNAs.

Thank you very much for this remark, which prompted us to come up with a better way to analyse the enrichment of factor binding among transcript classes. The new score now normalizes both by factor and by transcript class and is simply interpretable as a log enrichment score:

“For each factor, we counted the number of high confidence reads falling in each transcript class. […] The normalization can be interpreted as subtracting from the log enrichment matrix X’ the first singular component of its singular-value decomposition.”

Moreover, we have used non-unique mapping for this analysis to include duplicated genomic regions like rRNAs and tRNA isodecoders. The new procedure and its consequences for the analysis are described in our response to point 2 of reviewer 2.

5) "Dcp2, which harbors the hydrolase activity that removes the cap, cross-links frequently to mRNAs, in particular their 3 ´UTR, and to short noncoding transcripts, in especially to unstable SUTs. This suggests that Dcp2 is involved not only in cytosolic mRNA degradation but also in decapping during nuclear RNA surveillance." However, the defining characteristic that differentiates SUTs from CUTs is that SUTs are degraded by cytoplasmic Dcp2-dependent pathways as shown by several different labs (e.g. Pubmed IDs 17074811, 21610164, 21826286, 24931603). Thus rather than suggesting a new role for Dcp2, these findings confirm a different known role.

Thanks for pointing this out. We have revised the text accordingly:

“Whereas they [all decapping-related factors] show slight enrichment in binding to mRNAs, in particular to CDS and 3´ UTR, they preferentially bind to SUTs. This is consistent with previous findings that SUTs get degraded via Dcp2-dependent pathways in the cytoplasm (Marquardt et al., 2011; Smith et al., 2014; Thompson and Parker, 2007).”

6) "The analysis recovers established interactions between subunits of known complexes" but this is not supported by any quantitative analysis. Among the factors the authors analyze, some proteins that are thought to form stable complexes indeed cluster (e.g. Dcp1/2, Pan2/3) while other known stable complexes do not (e.g. Ski2/3/8 or the Air1 subunit of TRAMP).

The new replicate data has led to a generally better agreement between known protein complexes and clusters in our co-occupancy map (Figure 5C), such as the decapping and deadenylation complexes. We have updated the Results part to reflect this:

“The analysis recovers several established interactions between subunits of known complexes and between different complexes, providing a positive control. For example, all factors of the decapping complex show very high co-occupancy and co-localization, as do Air2 and Mtr4, which reside in the TRAMP4 complex.”

However, known complexes do not always correspond to clusters, as highlighted in the Discussion section:

“Although our data reflect factor cross-linking signal and measure occupancy on transcripts, and do not directly reveal function, the correlations of occupancies between factors and with transcript properties indicate functional aspects and suggest functional associations between factors that may guide future studies.”

7) Some of the names of the clusters in Figure 4C are poorly justified. For example, out of the six proteins in the "nuclear ncRNA processing" cluster four are cytoplasmic proteins that act mostly or exclusively on mRNA (Ccr4, Upf1, Ski3 and Ski7). They also function in degradation, rather than processing, thus at least the first two but possibly all three words in this name should be changed.

With the new datasets, clusters have gotten more in line with prior expectations. We have also tried to rename the clusters to reflect the biological functions more clearly (Figure 5C). However, we still see some factors amongst clusters that do not reflect the general function of the cluster. These are important findings that can be followed up by the community in the future. We highlighted and related unexpected findings in the text, e.g.:

“Factors involved in nuclear and cytoplasmic exosomal degradation (Rrp6, Csl4, Rrp4, Rrp40 and Rrp44) form a third cluster (cluster 3). […] Rai1 has been shown to detect and remove incomplete 5´ cap structures, to subject aberrant pre-mRNAs to nuclear degradation (Jiao et al., 2010).”

8) "For decapping factors (Dcp1, Edc3, Edc2, and Dhh1) and Xrn1, low codon optimality is the most determining feature for binding (Figure 5C).” However Figure 5C appears to show that this is true for three of these factors, but the opposite is true for Edc2, where high codon optimality appears to be determining. The authors should clarify.

Thank you for pointing this out. We generated two new replicates for Edc2 and those now show the same trends as other decapping factors with low codon optimality as the most determining feature for RNA binding (Figure 6—figure supplement 3). We do not know the reasons why Edc2 was an outlier in the previous data.

9) "Moreover, A-rich 4-mers are abundant around the proximity ( ± 8 nt) of Dcp2-cross-link sites (Figure 6C)." However Figure 6C and the legend say the window is ± 5 instead of ± 8. This might be a typo.

The correct range is ± 8. This typo was fixed in the manuscript.

10) "As expected, most PAR-CLIP cross-link sites fall into the mRNA transcript class, although about one third of the factors also show a considerable number of cross-links to non-coding RNAs". I suggest deleting "as expected" or replacing it with "surprisingly". Several of the factors have known roles in ncRNA processing/degradation and mRNA is only a minor fraction of cellular RNA and therefore I would not have expected most sites to fall into the mRNA class.A related issue may be that the analysis in Figure 1 appears to only include "Fractions of high-confidence PAR-CLIP cross-link sites" and not the actual number of or fraction of high-confidence reads. e.g. if there are 9 sites, in mRNA with 100 reads each and one in a ncRNA with 900 reads this would count as 90% of the sites being in mRNA, but only 50% of the reads are in mRNA. Since the complexity of mRNA is probably the highest (more than half the genome), the number of distinct sites in mRNA for a given factor is probably a systematic overrepresentation of that factor's importance in mRNA processing.

Thank you for this important comment. We included read distribution over annotated regions in Figure 2A and noted the change in the text:

“A first analysis revealed that most PAR-CLIP sequencing reads fall into the mRNA transcript class, although many of the factors also show a considerable number of sequencing reads in non-coding RNAs, in particular rRNA (Figure 2A).”

11) In Materials and methods "various non-overlapping transcript classes c" needs to be clarified. Several of the transcript classes in fact do overlap in the genome. Many snoRNAs are encoded within introns. Many CUTs, SUTs and NUTs overlap with mRNAs and each other. The methods need to be clarified.

This is indeed correct. We clarified how we retrieve our non-overlapping set in the Materials and methods section:

“We removed overlapping annotations with the following priority list: rRNA, tRNA, snRNA, snoRNA, intron, CDs, UTR, SUT, CUT, NUT.”

Reviewer #2:Specific points:1) Subsection “Transcriptome maps for 30 RNA degradation factors”: It appears that biological replicates were performed for only eleven out of the 30 degradation factors tested. For 19 of the factors analyzed, the conclusions are seemingly based on single experiments. While it is the case that the experiments reported involve a lot of work, it is quite unusual for the bulk of data in a manuscript to be published without clearer evidence of reproducibility. It would have been better to analyze fewer proteins in more depth. The authors present data indicating the consistency of the results from the proteins for which replicates were performed. However, while the replicates appear to cluster well in the 2D diagrams, inspection of the metagene analysis reveals appreciable differences between duplicates (Figure 1—figure supplement 1), which would presumably be much more marked at the level of individual genes.This is particularly an issue when the results are not as expected, e.g. the apparent binding of the deadenylase Ccr4 to a high proportion of tRNAs, which is based only a total of only 4,540 confident Ccr4 cross-linking sites.

Thank you very much for this important comment. We took this very seriously, collected many new datasets, and applied a more conservative threshold to eliminate weak old datasets. We are certain that only high-quality, highly reproducible data are included in the manuscript. Here is the reply to the same issue, raised by reviewer 1, point 1:

To ensure the highest possible data quality and reproducibility, we generated 49 new datasets, obtaining between two and four highly reproducible biological replicates for each degradation factor (Figure 1—figure supplement 1). Upon comparing the replicates, we found poor reproducibility for Pop2, Not1, Rrp4 and Caf40 and therefore omitted old replicates from the analysis. Furthermore, we applied a more conservative threshold for including data and eliminated six of the previous replicates that had less than 10.000 cross-link sites (Csl4, Rrp40, Rrp6, Ski2, Ski3, and Ski7). As a result, only highly reproducible data with a high number of crosslink sites were included in the final data set. This has led to a clearly visible improvement of the analysis and revised manuscript.

We now write:

“The high reproducibility of these PAR-CLIP experiments is revealed by a comparison of two independent biological replicates that we collected for all 30 degradation factors (Figure 1—figure supplement 1), with Spearman correlations between 0.87 and 1.00 (mean: 0.94).”

The extent of improvement of the PAR-CLIP data can be appreciated by comparing the old Figure 2 with new Figure 3.

2) The authors report that "most PAR-CLIP cross-link sites fall into the mRNA transcript class". This is unexpected, particularly for pre-rRNA processing factors, including the exosome and Mtr4, that should recover high level of rRNA hits. Inspection of the methods indicates that the reads were mapped to a unique database. This would have the effect of eliminating most rRNA and tRNA reads, as well as a subset of mRNAs. In the yeast genomic database, there are 2 copies of the rDNA repeat, so unique mapping will eliminate most rRNA hits. For tRNAs the situation is more complex; most tRNAs are present in multigene families and will be eliminated from unique hits, whereas the few single copy tRNA genes will be retained. This might be a particular issue for the reported Ccr4-tRNA interactions. For mRNAs, duplicated genes such as R-proteins will also be excluded.This point should be clarified. If uniquely-mapped reads were indeed used, the full mapping data should be presented.

Again, many thanks for this important observation. Unique mapping indeed contributed to low rRNA reads. However, since non-unique mapping negatively impacts the quality of cross-link sites in mRNA regions, we used two mapping strategies in the manuscript for Figure 2 and the rest of presented data (Materials and methods):

“We used two alternative approaches for mapping reads using Bowtie (Langmead et al., 2009): For all analyses except the ‘transcript class enrichment analysis’ in Figure 2, reads are uniquely mapped with up to one mismatch. […] For Figure 2, we therefore allowed Bowtie multi-mapping in two regions with –best, –starra options and discarded reads shorter than 30 nt."

This approach resulted in many more mapped reads in rRNA regions for Figure 2:

“A first analysis revealed that most PAR-CLIP sequencing reads fall into the mRNA transcript class, although many of the factors also show a considerable number of sequencing reads in non-coding RNAs, in particular rRNA (Figure 2A).”

3) Subsection “RNA end-processing complexes differ in their targets”: SUTs can be cytosolic, so this might not be sufficient to show that Dcp2 has nuclear activity.

We have changed the text to better reflect this point:

“For all decapping-related factors we observed similar binding preferences. […] This is consistent with previous findings that SUTs get degraded via Dcp2-dependent pathways in the cytoplasm (Marquardt et al., 2011; Smith et al., 2014; Thompson and Parker, 2007).

4) Subsection “Distinct factor distribution along mRNA”, Figure 1B (and end of subsection “PAR-CLIP experiments of S. cerevisiae proteins”): It would be helpful to discuss more why Csl4, Rrp4 and Rrp40 have distinct profiles. In particular, the Rrp4 profiles looks very different from all other degradation factors. This raises question of the PAR-CLIP experiment really worked for each cofactor. It is notable that some of these have low numbers of XL sites (561 for Rrp6, 923 for Rrp40, 7,373 for Csl4, while Rrp44 has much more 44,721).

We have repeated the experiments for members of the exosome to obtain 2-4 biological replicates with high reproducibility. In our new data-set we have at least 175,000 cross-link sites for each of the core exosome subunits and the metagene profiles show a consistent trend between them (Figure 3).

5) Subsection “Interactions between RNA processing machineries”, Figure 4 and Figure 4—figure supplements 1-2Figure 4: It would be helpful to discuss why Trf4/Air2 is bound to introns (Figure 1) but not Trf5 while Trf5/Air1 is showing higher co-occupancy / colocalization with splicing factors

With our new definition of enrichment (taking into account both row and column normalization), we see a stronger enrichment for Mtr4, Air1, and Air2 for intronic regions. This is reflected in the colocalization analysis (Figure 5—figure supplement 2), where both TRAMP4 and 5 show high correlation with splicing factors. In contract, co-occupancy reflects the tendency to bind to the same transcripts regardless of the binding position. Because TRAMP4 and 5 bind to most mRNAs, not just intron containing ones, its low co-occupancy with splicing factors is expected.

Reviewer #3:[…] The presented data are certainly of interest to a wide audience. However, there is no independent validation of any of the unexpected binding patterns. This is especially problematic, given that many datasets are based on single biological replicates. Moreover, some observations challenge our current thinking about the function of the investigated factor(s), which, to be suitable for publication in eLife, should be supported by independent experiments.

Thank you very much for this important comment. Here is the reply to the same issue, raised by reviewer 1, point 1:

To ensure the highest possible data quality and reproducibility, we generated 49 new datasets, obtaining between two and four highly reproducible biological replicates for each degradation factor (Figure 1—figure supplement 1). Upon comparing the replicates, we found poor reproducibility for Pop2, Not1, Rrp4 and Caf40 and therefore omitted old replicates from the analysis. Furthermore, we applied a more conservative threshold for including data and eliminated six of the previous replicates that had less than 10.000 cross-link sites (Csl4, Rrp40, Rrp6, Ski2, Ski3, and Ski7). As a result, only highly reproducible data with a high number of crosslink sites were included in the final data set. This has led to a clearly visible improvement of the analysis and revised manuscript.

We now write:

“The high reproducibility of these PAR-CLIP experiments is revealed by a comparison of two independent biological replicates that we collected for all 30 degradation factors (Figure 1—figure supplement 1), with Spearman correlations between 0.87 and 1.00 (mean: 0.94).”

The extent of improvement of the PAR-CLIP data can be appreciated by comparing the old Figure 2 with new Figure 3.

- Functional relevance:The manuscript presents a series of unexpected and somewhat controversial observations. One example concerns a possible role for Pop2 and Not1 in decay of CUTs. Any functional relevance of such binding should be trivial to validate. This would be relevant for other of the most striking findings. Such validation may be possible by re-analysis of published datasets, for example by intersecting occupancy with RNAseq upon depletion/mutation of the same factors.

Thank you for the comment. We have generated replicates and sometimes triplicates to show reproducibility of our data and significance of our observations. We observed problems with reliability of some factors in the older dataset, particularly Not1 and Pop2. We repeated the experiments for these factors at least twice. According to our new dataset, Pop2 and Not1 behave similarly to other members of the deadenylation machinery (Figure 2-4).

- Replicates and Technical artefacts:

*PAR-CLIP is a complex technique, where technical artefacts are not uncommon. For example, binding to highly abundant tRNA, snRNA and snoRNA species is often observed, even for factors where such binding is unlikely to occur or be relevant* in vivo*. Hence, it is quite possible that binding to such species, especially when based on a single biological replicate, does not reflect biology. Such considerations should be mentioned in the manuscript. Ideally, all factors for which single replicates are presented, should be backed up by additional data. This may be simply an additional replicate, or some kind of independent validation.*

More generally, more information on how the authors scrutinize the quality of their PAR-CLIP experiments should be given (i.e. more explicitly than simply referring to another publication).

Thank you for the comment. We have generated replicates and sometimes triplicates to show reproducibility of our data and significance of our observations. We observed problems with reliability of some factors in the older dataset, particularly Not1 and Pop2. We repeated the experiments for these factors at least twice. According to our new dataset, Pop2 and Not1 behave similarly to other members of the deadenylation machinery (Figure 2-4).

[Editors' note: the author responses to the re-review follow.]The manuscript has been considerably improved but there are some remaining issues that need to be addressed before acceptance, as outlined below:1) Even though some of the more controversial suggestions of the original draft are now resolved, comparison with available knock-out or mutant / RNA-seq datasets would greatly help to support, or dismiss, some of the mentioned novel findings.

While there are some RNA-seq experiments available to investigate functions for some of these factors, they were performed with different strains and under different conditions. However, we previously published a dataset of microarray expression measurements for the same strain background. In this work, synthesis and degradation rates upon knock-out of several degradation factors have been determined (Sun et al., 2013).

We correlated the PAR-CLIP-measured occupancy across mRNA transcripts with the change in degradation rate upon knock-out from this dataset. The heat map shows the Pearson correlations.

As expected, these values are anti-correlated, meaning that transcripts that are bound more by degradation factors experience a decrease in degradation rate upon knock-out. But unfortunately, we see little structure in the correlation matrix that would connect a PAR-CLIP profile to its respective knock-out experiment. Interestingly, we observed strongest anti-correlation of degradation rates with other degradation factor knock-outs for the deadenylation factor Pop2 with deadenylation and exosome components and the 5’→3’ exonuclease Xrn1 with decapping and NMD factors. This indicates a function for Pop2 in RNA degradation from the 3’ end, and an effect on RNA degradation from the 5’ end for Xrn1. We have previously shown a Xrn1 dependent buffering of transcript levels (Sun et al., 2013).

We believe this rather unstructured pattern is largely due to secondary effects based on the extensive buffering that balances changes in degradation rates by adapting synthesis rates when one of the factors is knocked-out (Sun et al., 2013). Moreover, the viability of these knock-out strains suggests that such bound transcripts could get processed by alternative degradation pathways. Finally, the lack of data on non-coding transcripts makes it difficult to validate further hypotheses. We have therefore not included this analysis in the manuscript.

2) If available, western blots and/or autoradiographs of the different PAR-CLIP experiments would be nice to include to allow readers to judge the quality of different IP experiments.

We included one representative Western Blot of the immunoprecipitation for each factor to show the quality of the different experiments as Figure 1—figure supplement 2.

3) Some of the conclusions presented in the Results sections still appear vague or unclear. This concerns for example interpretation of the RNA exosome binding profiles.a) i.e. In the first paragraph of the subsection “The exosome and surveillance factors”: unclear what different binding patterns "reflect" the distinct functions of the RNA exosome.

Thank you for pointing this out. We have changed the text to make the conclusions clearer: “This complex distribution of cross-links for different exosome subunits to different RNA classes reflects the distinct functions of the exosome in nuclear RNA surveillance, processing of stable non-coding RNAs, and cytoplasmic mRNA degradation (Zinder and Lima, 2017).”

b) In the second paragraph of the subsection “Distinct factor distribution along mRNA”: How/why would TRAMP4 binding upstream of the TSS affect early termination of the mRNA? Isn't it more plausible that it serves a role in termination of (unannotated) upstream ncRNA? Similarly, 5' bias of exosome and TRAMP subunits could arise from unannotated NUT-like transcripts.

Many thanks for pointing this out. Prompted by point 4 – as explained below – we have shown better now that the binding of TRAMP4 to regions upstream of TSS originates from NUT-like transcripts (compare Schulz et al., 2013). Since this explains the signal more readily, we have substituted the analysis with the one concerning attenuated genes. We have adapted the text in the second paragraph of the subsection “Surveillance of aberrant nuclear ncRNA”.

We also observed that upon removal of the cross-link sites that fall into previously annotated NUTs, and CUTs, the 5’ bias of the exosome remains intact (see new Figure 3—figure supplement 3). This suggests that such transcripts are apparently not the source of the binding bias on the 5’ end for exosome factors.

4) Figure 3 and Figure 3—figure supplement 1B: What RNAs generate the high signals for Mtr4 and Air2 5' to the TSS? This region is expected to give rise to very few transcripts. The upstream ncRNAs are further upstream and on the opposite strand. There must be a concern that some sequences have been mis-mapped. Were the sequences mapped only to a small number of loci, which would strengthen this concern? The authors should re-check the data supporting these distributions.

Thank you for this word of caution, which prompted us to investigate in more detail the origins of this signal. The reads are mapped uniquely to the whole genome with a minimum length of 20 bases and just one mismatch allowed. This strict mapping criteria makes it very unlikely for reads to get mis-mapped. However, we showed before that upon Nrd1 knock-out aberrant transcription upstream of TSS becomes visible (Schulz et al., 2013). To show that the responsible Nrd1-unterminated transcripts (NUTs) are indeed the RNAs for which we see TRAMP4 binding signal, we performed two further analyses that are explained in the text:

“It has been shown that Nrd1 is involved in terminating transcripts upstream of the TSS. […] Comparison of the observed antisense profiles (Figure 4) with those obtained after excluding cross-link sites in previously annotated NUT and CUT regions (Figure 4—figure supplement 2) confirms that most of the signal originates from transcripts that are targeted by the Nrd1/Nab3 machinery.”

5) The second paragraph of the subsection “The exosome and surveillance factors” refers to Figure 2—figure supplement 3 and not Figure 3—figure supplement 3.

Thank you. We corrected the text accordingly.

6) Figure 5C: 3' degaradation => 3' degradation

Thank you. We corrected the figure accordingly.